# Towards Universal Mesh Movement Networks

**Mingrui Zhang**[1]    **Chunyang Wang**[1]    **Stephan Kramer**[1]    **Joseph G. Wallwork**[2]
**Siyi Li**[1]    **Jiancheng Liu**    **Xiang Chen**[3]    **Matthew D. Piggott**[1]
[1]Imperial College London    [2]University of Cambridge    [3]Noah's Ark Lab, Huawei
{mingrui.zhang18,chunyang.wang22,s.kramer}@imperial.ac.uk
{siyi.li20,m.d.piggott}@imperial.ac.uk
jw2423@cam.ac.uk  xiangchen.ai@outlook.com  ljcc0930@gmail.com

## Abstract

Solving complex Partial Differential Equations (PDEs) accurately and efficiently is an essential and challenging problem in all scientific and engineering disciplines. Mesh movement methods provide the capability to improve the accuracy of the numerical solution without increasing the overall mesh degree of freedom count. Conventional sophisticated mesh movement methods are extremely expensive and struggle to handle scenarios with complex boundary geometries. However, existing learning-based methods require re-training from scratch given a different PDE type or boundary geometry, which limits their applicability, and also often suffer from robustness issues in the form of inverted elements. In this paper, we introduce the Universal Mesh Movement Network (UM2N), which – once trained – can be applied in a non-intrusive, zero-shot manner to move meshes with different size distributions and structures, for solvers applicable to different PDE types and boundary geometries. UM2N consists of a Graph Transformer (GT) encoder for extracting features and a Graph Attention Network (GAT) based decoder for moving the mesh. We evaluate our method on advection and Navier-Stokes based examples, as well as a real-world tsunami simulation case. Our method out-performs existing learning-based mesh movement methods in terms of the benchmarks described above. In comparison to the conventional sophisticated Monge-Ampère PDE-solver based method, our approach not only significantly accelerates mesh movement, but also proves effective in scenarios where the conventional method fails. Our project page can be found at `https://erizmr.github.io/UM2N/`.

## 1   Introduction

Various natural phenomena are modeled by Partial Differential Equations (PDE). The accurate and efficient approximation of the solutions to these complex and often nonlinear equations represents a fundamental challenge across all scientific and engineering disciplines. To solve real-world PDEs, many numerical methods require a computational mesh or grid to discretize the spatial domain. The quality of this mesh significantly impacts the balance between a numerical solution's accuracy and the computational cost required to obtain it. To maintain high-resolution everywhere in the domain is computationally expensive. Many systems modeled by PDEs however are multi-resolution in nature. For example, a small part of the system may be highly dynamic, while other regions are quasi-stationary; alternatively, some locations may be more important while others less important to the question being considered by the calculation [1–4]. While a uniform, high-resolution mesh can be utilized to capture the dynamics or what is most important accurately, this is often wasteful of finite, precious computational resources and comes with sustainability implications [5].

An active body of research is focused on developing deep learning based methods to accelerate the PDE-solving process by learning surrogate neural solvers [6–8]. A learned solver can output PDE

solutions directly given the physical parameters, and boundary conditions of a PDE. However, these methods encounter challenges such as not being able to guarantee physical plausibility of the PDE solution, weak generalization ability, and low data efficiency.

An alternative approach to improve the efficiency of a PDE solver is to utilize *mesh adaptation*, which is a technique for distributing the mesh according to numerical accuracy requirements. Two main categories of mesh adaptation techniques can be identified: $h$-adaptation and $r$-adaptation. $h$-adaptation refines or coarsens the mesh resolution dynamically through topological operations such as adding/deleting nodes and swapping element edges/faces. In contrast, $r$-adaptation (or *mesh movement*) relocates or moves mesh nodes, keeping the mesh topology and thus the number of elements and vertices in the mesh unchanged [9]. These traditional mesh adaptation techniques can help reduce PDE-solving costs, but the mesh adaptation process itself may come at the cost of significant computational overhead.

Deep learning based methods have been proposed to accelerate mesh adaptation. [10–16]. Most previous methods focus on $h$-adaptive refinement [11–13, 17]. Some existing works focus on learning error indicators or Riemannian metrics (which account for element shape and anisotropy, as well as size) [13–16] to guide the mesh adaptation process. The learned indicator or metric is then fed into a traditional remesher for mesh generation, or a mesh optimizer, which overall limits the performance of these works to being no better than traditional methods, especially in terms of efficiency. Reinforcement learning based methods show potential to improve the mesh adaptation task, but are difficult to train with low data efficiency [11, 17, 18]. Only a small number of works focus on $r$-adaptation. [19, 20] investigated supervised and unsupervised learning based mesh movement methods. However, the proposed methods require re-training from scratch given a different PDE or geometry, which limits their applicability. In addition, the unsupervised learning method requires significant GPU memory and suffers from long training times, which makes re-training even more prohibitive for large scale problems.

Targeting the generalization ability and efficiency of mesh movement, here we introduce the Universal Mesh Movement Network (UM2N), a two stage, deep-learning based model that learns to move the computational mesh guided by an optimal-transport approach driven by solutions of equations of Monge-Ampère (MA) type. Given an underlying PDE to solve numerically, the process of MA based mesh movement includes choosing a suitable monitor function (*i.e.*, a measure for the desired mesh density) and solving an auxiliary PDE whose solution prescribes the movement of the mesh. The Monge-Ampère equation is an example of such an auxiliary PDE, which results in mesh movement with attractive theoretical properties [9], but whose solution comes at a significant computational cost. Therefore, we decouple the underlying PDE solve and the mesh movement process. The learning of the mesh movement process is essentially learning to solve the auxiliary Monge-Ampère equation.

The proposed UM2N consists of a Graph Transformer based encoder and a Graph Attention Network (GAT) [21] based decoder.

Element volume loss is selected within the training loss, instead of the coordinates loss used in previous works [19, 20]. This offers the advantage that element volume loss not only provides supervision signals but also penalizes negative volumes (i.e., inverted elements), thereby reducing mesh tangling and enhancing robustness. We construct a PDE-independent dataset by generating random generic fields for model training. The trained model can be applied in a zero-shot manner to move meshes with different sizes and structure, for solvers applicable to different PDE types and boundary geometries. We comprehensively demonstrate the effectiveness of our methods on different cases including advection, Navier-Stokes examples, as well as a real-world tsunami simulations on meshes with different sizes and structures. Our UM2N outperforms existing learning-based mesh movement methods in terms of the benchmarks described above. In comparison to the sophisticated Monge-Ampère PDE-solver based method, our approach not only significantly accelerates mesh movement, but also proves effective in scenarios where the conventional method fails. The trained model can be directly integrated into any mesh based numerical solvers for error reduction and acceleration in a non-intrusive way, which can benefit various engineering applications suffering trade-offs between accuracy and computational cost.

Our main contributions are listed as follows:

- We present the UM2N framework, which once trained can be applied in a non-intrusive, zero-shot manner to move meshes with different sizes and structures, for solvers applicable to different PDE types and boundary geometries.

- We propose PDE-independent training strategies and mesh tangling aware loss functions for our Graph Transformer and GAT-based architecture.

- We demonstrate the universal ability of UM2N on different PDEs and boundary geometries: Advection, Navier-Stokes examples, and a real-world tsunami simulation case.

## 2   Related Works

**Neural PDE solver.** Neural PDE surrogate models can significantly accelerate the PDE solving process. There are three main groups of existing works. Physics-informed Neural Networks (PINNs) [6, 22, 23] are the first group, modeling the PDEs through implicit representations by neural networks. PINNs require knowledge of the governing equations and train neural networks with equation residuals from the PDEs with given boundary conditions and initial conditions. Neural operators are another group of methods for learning to solve PDEs which seek to learn a mapping from a function describing the problem to the corresponding solution function [7, 8, 24–26]. The third group is mesh-based PDE solvers with deep learning, including the utilization of CNNs and GNNs for processing structured and unstructured meshes [15, 27–32]. Our method can be considered to be related to the neural operator approach, although it learns a mapping from monitor function values to the potential field describing the mesh movement instead of the PDE solution directly. In addition, the networks in the proposed model share similar ideas to mesh-based PDE solvers for processing unstructured meshes.

**Learning for mesh generation and adaptation.** Deep learning methods have been advanced for applications in mesh generation and adaptation. Two tasks need to be differentiated here, the first involves modeling the ideal mesh resolution or density, *e.g.* through learning a metric or error indicator. The second involves the step that actually adapts the mesh in response, where we can distinguish between $h$-adaptivity, in which the connectivity structure of the mesh is changed to refine or coarsen the mesh, and $r$-adaptivity which maintains the connectivity and only moves the vertices.

An example of the first task is MeshingNet [10] which employs a neural network to establish the requisite local mesh density, subsequently utilized by a standard Delaunay triangulation-based mesh generator. Similarly, the determination of optimal local mesh density for the purpose of mesh refinement is described in [12]. In [13], the authors propose a learning-based model to determine optimal anisotropy to guide $h$-adaptation. [15] introduces a neural-network-based approach for predicting the sizing field for use with an external remesher. Instead of learning a metric, some works learn to manipulate the mesh directly. [11, 17, 18] formulate the $h$-adaptation process as a reinforcement learning paradigm, learning a policy that guides to adjust the mesh elements.

$r$-adaptation (or mesh movement) is generally formulated in a fundamentally different manner to $h$-adaptation. Leading $r$-adaptation methods are based on a mapping between a fixed reference computational domain and the physical domain encompassing the adapted mesh, which is established through the solution of a nonlinear auxiliary PDE. [19] first proposed the use of a GNN based mesh movement network. [20] investigated the unsupervised learning of mesh movement for improving performance of neural PDEs solvers. [33] introduced a learning based mesh movement method specially tailored for computation fluid dynamics simulation of airfoil design.

Our work focuses specifically on $r$-adaptation, *i.e.*, mesh movement methods. The approaches proposed in previous work require re-training the model when applied to different problems, or can only generalize within a limited range of PDE parameters. We propose a mesh movement network that aims to be universal across applications to different PDEs without retraining.

## 3   Preliminaries and Problem Statement

**Monge-Ampère mesh movement.** A mesh movement problem can be defined as the search for the transformation between a fixed computational domain $\Omega_C$, and a physical domain $\Omega_P$. Coordinates in these spaces are denoted $\boldsymbol{\xi}$ and $\mathbf{x}$, respectively. The mesh $\mathcal{H}_C$ defined in the computational domain is the original, often uniform mesh. The mesh $\mathcal{H}_P$ defined in the physical domain is the

adapted (or moved) mesh.[1] The goal of mesh movement is to find a mapping $\boldsymbol{x} = f(\boldsymbol{\xi})$ between the continuous coordinate fields of $\Omega_C$ and $\Omega_P$, although in practice we deduce a mapping of the discrete vertex sets of $\mathcal{H}_C$ and $\mathcal{H}_P$. While the coordinate transformation can also be used to accommodate time-dependent moving boundaries [34] or the optimisation of the shape of the domain, here we focus on its application to achieve variable resolution such that the solution $u_{\boldsymbol{x}}$ of a PDE solved on the adapted mesh has higher accuracy than the solution $u_{\boldsymbol{\xi}}$ solved on the original mesh.

A monitor function $m$ over the spatial domain is used to specify the desired mesh density, which can be based on various characteristics of the solution field $u$ of the PDE. It prescribes where the mesh should be refined or coarsened, *i.e.* large monitor values indicate where high mesh resolution is required. Therefore, the goal of the mesh movement process can be rephrased as finding a mapping so that $m$ is equidistributed over the adapted (*i.e.*, physical) mesh [35]:

$$m(\boldsymbol{x}) \ \det(\boldsymbol{J}) = \theta, \tag{1}$$

where $\boldsymbol{J}$ is the Jacobian of the map $\boldsymbol{x} = f(\boldsymbol{\xi})$ with respect to the computational coordinates $\boldsymbol{\xi}$, $\det(\boldsymbol{J})$ denotes the determinant of the Jacobian which corresponds to the relative change in volume under the transformation, and $\theta$ is a normalization constant. By using concepts from optimal transport theory [36], the problem can be constrained to have a unique solution, with the deformation of the map expressed in terms of the gradient of a scalar potential $\phi$ such that:

$$\boldsymbol{x}(\boldsymbol{\xi}) = \boldsymbol{\xi} + \nabla_{\boldsymbol{\xi}}\phi(\boldsymbol{\xi}). \tag{2}$$

Substituting the additional constraint described by equation (2) into equation (1) gives a nonlinear PDE of Monge-Ampère type:

$$m(\boldsymbol{x}) \ \det(\boldsymbol{I} + \boldsymbol{H}(\phi(\boldsymbol{\xi}))) = \theta, \tag{3}$$

where $\boldsymbol{I}$ is the identity matrix and $\boldsymbol{H}(\phi)$ is the Hessian of $\phi$, with derivatives taken with respect to $\boldsymbol{\xi}$, i.e, $\boldsymbol{H}(\phi)_{ij} = \frac{\partial^2 \phi}{\partial \xi_i \partial \xi_j}$. It should be noted that guarantees for existence and uniqueness, and convergence of solution methods for the Monge-Ampère equation generally rely on the domain being diffeomorphic with a convex domain (e.g. [37]). In practice, the method is known to break down in some cases where the domain is not simply-connected or has non-smooth boundaries.

The MA equation (3) is an auxiliary PDE which is purely associated with the the mesh movement process, and is independent of the underlying PDE or physical problem we wish to solve, i.e., the MA equation keeps the same form for different PDEs. This decoupling benefits learning-based methods in terms of generalization properties, *i.e.*, a well-learned MA neural solver has the potential to be applied to different problems with a small cost of fine-tuning or even without re-training.

## 4 UM2N: Universal Mesh Movement Network

### 4.1 Framework overview.

The proposed UM2N framework is shown in Fig. 1. Given an input mesh, vertex features and edge features are collected separately. The coordinates and monitor function values are gathered from the vertices and input into a graph transformer to extract embeddings. The embedding vector $\boldsymbol{z}$ obtained from the graph transformer encoder is then concatenated with the extracted edge fea-

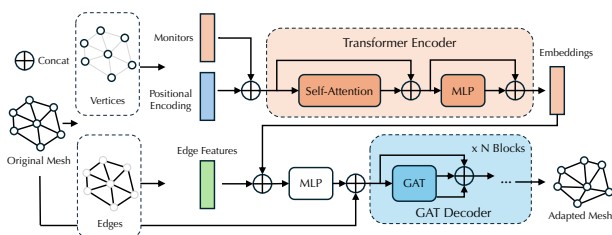

Figure 1: Overview of Universal Mesh Movement Network.

tures $\boldsymbol{e}$ to serve as the input for the Graph Attention Network (GAT) decoder. The decoder processes this combined input along with a mesh query to ultimately produce an adapted mesh.

**Graph Transformer encoder.** Transformers [38] are a popular architecture which have been successfully applied for neural PDE solvers [25, 26, 39]. They have strong expressivity and can naturally handle irregular data structures such as unstructured meshes. Their attention mechanism can capture both the local and global information for a vertex of interest in a mesh. Here we use a

---

[1]In what follows, we abuse notation slightly by also using $\boldsymbol{\xi}$ to refer the the (discrete) vertices of $\mathcal{H}_C$, rather than the (continuous) coordinate field. Similarly, we also use $\boldsymbol{x}$ to denote the vertices of $\mathcal{H}_P$.

graph transformer encoder as a feature extractor. The input features include the coordinates of the original mesh $\boldsymbol{\xi}$ and monitor values $m_{\boldsymbol{\xi}}$. The coordinates serve as the positional encoding. Features are concatenated and encoded into $\boldsymbol{Q}$ (query), $\boldsymbol{K}$ (key) and $\boldsymbol{V}$ (value) as inputs using MLPs for a self-attention module. The encoder output embedding $\boldsymbol{z}$ are fed into a downstream GAT based deformer for mesh movement.

**Graph Attention Network (GAT) based decoder.** We choose GATs to construct our decoder. Its attention mechanism can help constrain the mesh vertex movement to within one hop of its neighbors which assists in alleviation of mesh tangling issues. The decoder consists of $N$ GAT blocks. The first GAT block takes a mesh query $\boldsymbol{\xi}$, and the embedding $\boldsymbol{z}$ from the transformer encoder as well as the edge features $\boldsymbol{e}$ as inputs. For each following $k^{\text{th}}$ block, the inputs consist of the coordinates of the initial mesh $\boldsymbol{\xi}$, the coordinates of the intermediate moved mesh $\boldsymbol{\xi}^{(k-1)}$ and extracted features from the previous layer $\boldsymbol{f}^{(k-1)}$.

## 4.2 PDE-independent dataset and training.

In the existing work of neural mesh movement [19, 20], proposed models are trained on solutions of PDEs, with the training data generated by solving one specific type of PDE, limiting generalizability of the trained model. Aiming here to train universal mesh movement networks, we construct a PDE-independent training dataset $\mathcal{D} = \{\boldsymbol{d} = (\boldsymbol{\xi}, m_{\boldsymbol{\xi}}; \boldsymbol{V}_r, \boldsymbol{T}_r)\}$, where $\boldsymbol{\xi}, m_{\boldsymbol{\xi}}$ denotes the original mesh and monitor values as the model's input, and $\boldsymbol{V}_r, \boldsymbol{T}_r$ as the pre-calculated (reference) mesh's vertex coordinates and elements as ground truth. To build the dataset, we randomly generate generic solution fields, which are composed as the summation of a random number of Gaussian distribution functions centred in random locations, and with random widths in different directions to introduce anisotropy, samples are shown in Figure A1; refer to Appx. B for a detailed description. The generated fields can be interpreted as representing solutions of any PDE. For mesh movement, a widely used method to translate the PDE solution $u(\mathbf{x})$ into a monitor function, is the Hessian based formula [36]:

$$m(\mathbf{x}) = 1 + \alpha \frac{\|H(u)\|}{\max \|H(u)\|}, \tag{4}$$

where $\|H(u)\|$ indicates the Frobenius norm of the Hessian of $u$ and $\alpha$ is a user chosen constant. This choice results in small cells where the curvature of the solution is high, with a ratio of $1 + \alpha$ between the largest and smallest cell volumes. Our network is trained on the values of the monitor function only, which can be computed relatively cheaply from the PDE solution. The expected movement of the mesh vertices is computed by solving the Monge-Ampère equation (3) for each of the monitor functions based on the randomly generated PDE solutions. After training, the model can then be applied using only the monitor values as input, independent of the PDE that is being solved. Other choices of formulae for the monitor function in terms of $u$ may be appropriate in different cases, but again these can be applied without re-training the model, *e.g.*, the flow-past-a-cylinder and tsunami test cases in the following section use a monitor function based on the gradient of the solution.

## 4.3 Loss functions.

Given the dataset $\mathcal{D}$ defined above, the final objective is to find model parameters $\boldsymbol{\theta}$, by minimizing the total loss $L_{total}$. Thus, we can formalize the process as

$$\arg\min_{\boldsymbol{\theta}} L_{total}(\boldsymbol{\theta}) := \lambda_{vol} L_{vol}(\boldsymbol{\theta}) + \lambda_{cd} L_{cd}(\boldsymbol{\theta}). \tag{5}$$

where $L_{vol}, L_{cd}$ represents the element volume loss and Chamfer distance loss respectively, which are defined later. $\lambda_{cd}, \lambda_{vol} > 0$ represent hyper-parameters balancing these two effects.

We denote the modified (adapted) mesh with model parameters $\boldsymbol{\theta}$ as $\mathcal{M}(\boldsymbol{\xi}, m_{\boldsymbol{\xi}}; \boldsymbol{\theta}) = \{\boldsymbol{V}_a, \boldsymbol{T}_a\}$, where $\boldsymbol{V}_a = \{\boldsymbol{x}_i\}_{i=1}^{n_1}$ represents the coordinates of vertices and $\boldsymbol{T}_a = \{\boldsymbol{t}_i\}_{i=1}^{n_2}$ represents the elements with $\boldsymbol{x}_i, \boldsymbol{t}_i$ as the $i^{\text{th}}$ vertex and element. For ease of presentation, we omit the $\boldsymbol{\theta}$ here. Similarly, we define $\boldsymbol{y}_i, \boldsymbol{q}_i$ as $\boldsymbol{V}_r, \boldsymbol{T}_r$'s $i^{\text{th}}$ vertex and element.

**Element volume loss.** In contrast to the coordinate loss used in previous work [19, 20], here we use the element volume loss for training. Element volume loss is computed as the averaged volume difference between each element in the adapted mesh ($\boldsymbol{T}_a$) and the reference mesh ($\boldsymbol{T}_r$) computed from the MA method. It is defined as

$$L_{vol}(\boldsymbol{\theta}) = \mathbb{E}_{(\boldsymbol{\xi}, m_{\boldsymbol{\xi}}; \boldsymbol{V}_r, \boldsymbol{T}_r) \in \mathcal{D}} \left[ \frac{1}{|\boldsymbol{T}_a|} \sum_{i=1}^{n_2} |\text{Vol}(\boldsymbol{t}_i) - \text{Vol}(\boldsymbol{q}_i)| \right], \tag{6}$$

where Vol is the function computing the volume of a given element $\boldsymbol{t}$.

The element volume loss is inspired by the equidistribution relation in equation (1). Intuitively, the equidistribution relation enforces deformed mesh elements such that their volumes correspond to rescaling by the monitor function value $m(\boldsymbol{x})$. Therefore, the element volume loss is utilised to encourage the model to learn to move the mesh so as to conform to the equidistribution relation, under the guidance of MA method, given that the MA method provides one good way to achieve this relation. In addition, the element volume loss penalizes negative Jacobian determinants, which helps prevent element inversion, *i.e.*, mesh tangling.

**Chamfer distance loss.** Chamfer distance finds for all vertices in the first mesh, the nearest vertex in a second mesh and sums the square of these distances. It encourages the model to output a mesh which has a similar spatial distribution of vertices to that of the reference mesh. To achieve vertex-to-vertex alignment, the bidirectional Chamfer distance is utilized without additional sampling. It is defined as

$$L_{cd}(\boldsymbol{\theta}) = \mathbb{E}_{(\boldsymbol{\xi}, m_{\boldsymbol{\xi}}; \boldsymbol{V}_r, \boldsymbol{T}_r) \in \mathcal{D}} \left[ \frac{1}{|\boldsymbol{V}_a|} \sum_{\boldsymbol{x}_i \in \boldsymbol{V}_a} \min_{\boldsymbol{y}_j \in \boldsymbol{V}_r} \|\boldsymbol{x}_i - \boldsymbol{y}_j\|_2 + \frac{1}{|\boldsymbol{V}_r|} \sum_{\boldsymbol{y}_j \in \boldsymbol{V}_r} \min_{\boldsymbol{x}_i \in \boldsymbol{V}_a} \|\boldsymbol{x}_i - \boldsymbol{y}_j\|_2 \right]. \tag{7}$$

## 5 Experiments

### 5.1 Experiment setups.

Unlike existing approaches to mesh movement, such as [19, 20], which re-train their model on a case-by-case basis, our study aims to explore the universal applicability of the proposed UM2N. Therefore, all examples shown in this section are tested in a zero-shot generalization manner, *i.e.*, without re-training. In addition to PDE type, our training meshes are of small size typically comprising 500 vertices, while our test meshes have far more vertices: Advection ($2,052$ vertices), Cylinder ($4,993$ vertices), Tsunami ($8,117$ vertices), which also tests the model's scalability. There are examples with more complicated settings, please refer to Appx. E.

**Training.** The training dataset consists of 600 randomly generated generic solution fields and original meshes with 463 and 513 vertices. The model is trained using the Adam optimizer. The training and experiments are performed on an Nvidia RTX 3090 GPU.

**Metrics.** The main metric for evaluating mesh movement quality is underlying PDE solution error reduction (ER) ratio. Here PDE error is approximated by the difference between the solutions obtained on a coarse mesh and accurate solutions which are obtained on a very high resolution mesh. The PDE error reduction ratio is the difference between the PDE error from an original mesh and that from an adapted mesh. Another important metric for mesh movement is mesh tangling. Once mesh tangling happens, the mesh is invalid for a numerical solver, which breaks the simulation.

### 5.2 Benchmarking mesh movement.

Table 1: Quantitative results across mesh movement methods. ER indicates PDE error reduction ratio. "Fail" indicates that mesh tangling happens during the simulation.

| Methods | Swirl | | Cylinder | | Helmholtz | |
|---|---|---|---|---|---|---|
| | ER (%) ↑ | Time (ms) ↓ | ER ↑ (%) | Time (ms) ↓ | ER (%) ↑ | Time (ms) ↓ |
| MA [36] | **37.43** | 5615.78 | Fail | - | **17.95** | 3265.23 |
| M2N [19] | 14.15 | 25.01 | Fail | - | 16.32 | **4.62** |
| UM2N (Ours) | 35.93 | **17.66** | 61.29 | 19.25 | 17.08 | 4.86 |

In the following section, we benchmark the mesh movement using both non-learned Monge-Ampère (MA) mesh movement implemented in *Movement* [40] and learning-based baseline M2N [19] over three different scenarios to compare with UM2N. Table 1 shows the quantitative results in three different scenarios, Swirl, Cylinder, and Helmholtz. An additional qualitative result of Tōhoku Tsunami simulation, is provided to show our model can handle highly complex boundaries in real-world scenarios. Moreover, to give a fair comparison, we re-trained the model in M2N on our training dataset before evaluation.

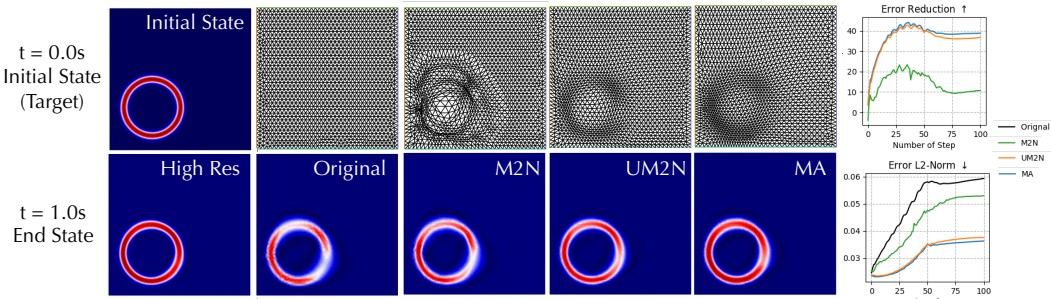

Figure 2: Results for swirl case. The plot shows the end state of the swirl simulation (bottom row) and corresponding meshes (top row). As the process is reversible, the end state should be consistent to the initial state. It can be observed that the solution obtained on high resolution almost recover the initial state. The proposed UM2N outperform M2N and achieve comparable performance to MA method. Quantitatively in the rightmost plot, the orange line (UM2N) and blue line (MA) best suppress error accumulation along timesteps.

**Helmholtz.** The Helmholtz case is a time-independent problem solving a 2D Helmholtz equation as in (A2). The source terms of the equation are generated use formula (A1). Our model achieves better error reduction than M2N and comparable to MA method in significantly less time.

**Swirl.** The swirl case considers a time-dependent pure advection equation. In this scenario, the initial tracer field takes a narrow ring shape and is advected within a swirling velocity field, see Eq. (A3) and (A4) in Appx. C. The velocity swirling direction is anti-clockwise in the first half of the simulation and clockwise for the second half. Given this setting, as the advection process is reversible, the final state should be consistent with the initial state. This constitutes a problem of significant multi-resolution complexity, as the extremely narrow ring requires a high resolution to accurately resolve the dynamics at the sharply defined interfaces of both inner and outer sides.

The results are shown in Fig. 2. As shown in Table 1, our UM2N shows an average $35.93\%$ error reduction which clearly outperforms M2N's $14.15\%$. It is close to that of the full Monge-Ampère (MA) mesh movement approach which achieves an error reduction of $37.43\%$, but with significantly lower computational time.

**Flow past cylinder.** Simulation of flow past a cylinder in a channel is a classic and challenging multi-resolution case of computational fluid dynamics, solving the Navier-Stokes equations. The cylinder and the channel's top-bottom surfaces require a high resolution mesh to resolve the boundary layers [41]. Given appropriate Reynolds number, the well-known von Kármán vortex street phenomena can be observed within the wake flow which also benefits from higher resolution to resolve the time-varying evolution of vorticity structures. The drag $C_D$ and lift $C_L$ coefficients for the cylinder are other quantities of interest in this problem. Here we use a benchmark configuration from [42], please refer to Appx. C.3 for more details.

The simulation is run for $8,000$ time steps of size $0.001$ s, *i.e.*, $8$ s physical time for the whole simulation. The original mesh is a uniform mesh with triangular elements and $4,993$ vertices. The reference solutions are obtained on a high resolution mesh with $260,610$ vertices.

We investigate the dynamics in the area around the cylinder and in the wake flow respectively. For the former, we use the measure of $C_D$ as shown in Fig. 3. The results for $C_L$ is shown in Fig. A10 in Appx. E. For the wake flow, we use cross-section probes at $x = [0.5, 1.0, 1.5, 2.0]$ to measure the intensity of vorticity spatially along the $y$-axis as shown in Fig. 4.

Qualitatively, it can be observed that the proposed method moves the mesh in a manner that captures the fluid dynamics for the whole domain in general (see upper part of Fig. 3). At the cylinder and top-bottom surface, the vertices are moved to increase the resolution in order to resolve the boundary layers as expected. Quantitatively, the drag $C_D$ and lift $C_L$ coefficients computed on our moved mesh show a $61.29\%$ average error reduction compared to that of original mesh as shown in lower part of Fig. 3. The proposed UM2N approach not only improves the accuracy of these coefficients in magnitude but also prevents their periodic variation from phase shifting. For the wake flow, as shown in Fig. 4, we selected a single snapshot at $t = 4.0s$ for qualitative investigation. Our UM2N also shows an accuracy improvement for vorticity intensity on data sampled from all four cross-section probes comparing to the that on original mesh.

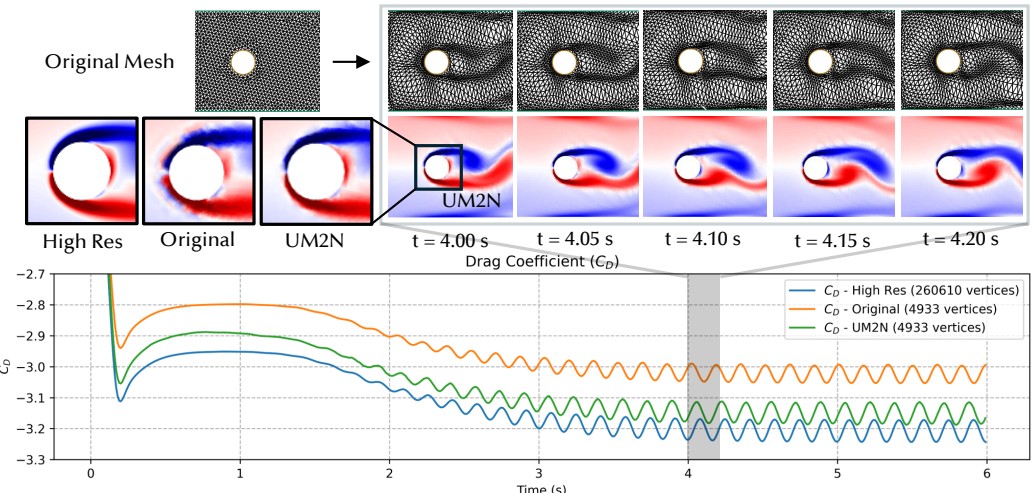

Figure 3: Results for the flow past cylinder case in time. In the upper part, we visualize 5 snapshots of adapted meshes and vorticity intensity from $4.00s$ to $4.20s$. Please refer to the full video in the supplementary materials. It shows that our UM2N can output meshes which adapt to the dynamics around the cylinder. It can be observed from the zoom-in mini-plots at the top-left that UM2N reduces errors/noise in vorticity intensity compared to the original mesh. In the lower part, the blue, orange and green lines show the drag $C_D$ coefficients obtained on a high resolution fixed mesh, UM2N adapted mesh and the original coarse mesh for the first 6 seconds (*i.e.*, 6000 steps). It can be observed that the UM2N output mesh improves the accuracy of $C_D$ in magnitude and periodicity compared to the original mesh.

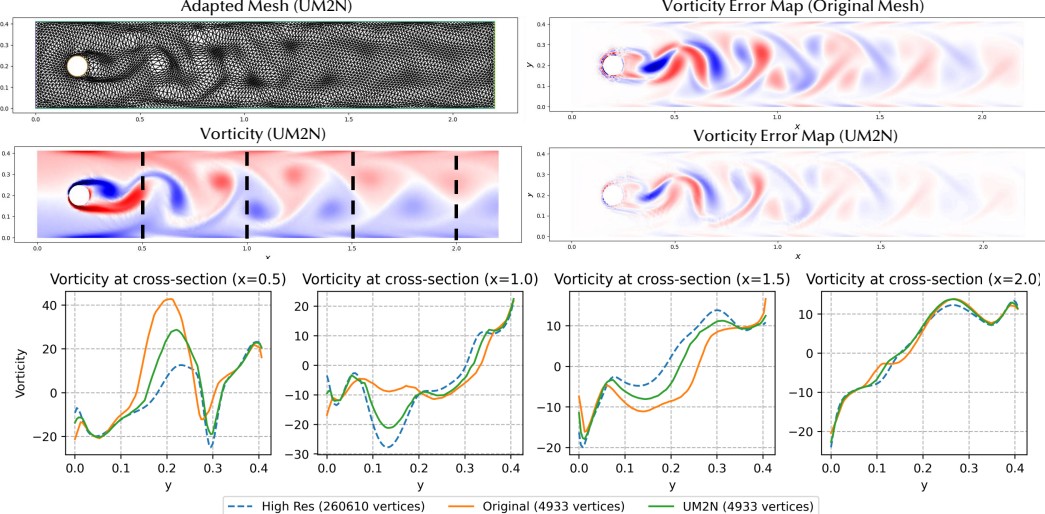

Figure 4: Analysis on wake flow vorticity for the cylinder case. The adapted mesh output by UM2N (see top-left part of the plot). Qualitatively, the error maps at the top-right part indicate that our UM2N reduce the vorticity difference to the high resolution mesh result in general. The quantitative comparison results along the probes are shown in the lower part of the figure. There are four cross-section probes place on $x = [0.5, 1.0, 1.5, 2.0]$ marked by the black dash line in the middle left plot. The green line (our UM2N) is much more consistent to the blue dash line (high resolution) compared to the orange line (original mesh), which further verifies that our method can reduce the PDE errors.

Note that we intended to compare our UM2N to M2N and the conventional full MA method. However, both MA and M2N fail *i.e.*, the solver diverges, within 30 time steps due to mesh tangling and thus provide insufficient data points for comparison. For comparisons with a non-tangled M2N case, please refer to the simplified laminar flow-past-cylinder example presented in Appendix D.1.

**Tōhoku tsunami simulation.** Here we present the case of a simulation of the 2011 Tōhoku Tsunami, to show that our methods can be applied to real world scenarios with boundaries describing highly complex geometries. The tsunami is simulated by solving the shallow water equations using the Thetis framework [43]. For details on the datasets and software used to set up the Tōhoku case study, see Appx. B.2. Note that the mesh (8, 117 vertices) has an arbitrarily irregular boundary. It can be

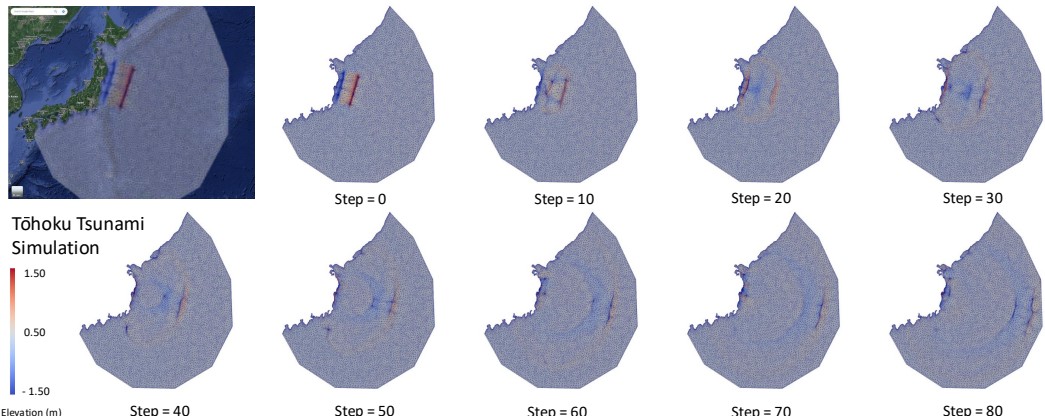

Figure 5: Qualitative results for the Tōhoku tsunami simulation. The boundary of the simulation is generated based on the real coastline data (see the mesh overlaid on the satellite map). We show 9 snapshots of the wave elevation from 80 steps of the tsunami simulation enhanced by our UM2N. In these visualizations, red and blue hues indicate wave elevations that are, respectively, above or below mean sea level. Regions exhibiting significant elevation magnitudes necessitate increased resolution to accurately resolve the underlying dynamics. Our method dynamically and robustly adjusts the mesh to increase resolution at the wave front, thereby effectively tracking its propagation.

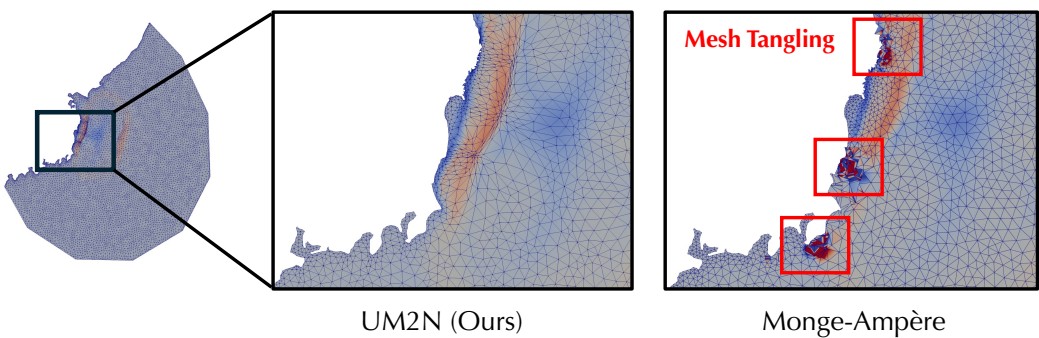

Figure 6: Handling complex boundaries. The Monge-Ampère method generates inverted elements, resulting in mesh tangling near the coastline as shown in the areas delineated by red rectangles. Our UM2N can smoothly handle the elements near the coastline without mesh tangling.

observed that the mesh moves in a manner that helps track the wave propagation with enhanced resolution, as shown in Fig. 5. Our model can also robustly handle the highly complex boundary without mesh tangling near the coastline. As a comparison, the conventional Monge-Ampère (MA) mesh movement approach struggles to handle the complex geometries, *i.e.*, fails to converge when solving the non-linear MA equation, resulting in highly tangled elements near the coastline as shown in Fig. 6. Therefore, our method not only provides a more efficient approach but also show advantages dealing with complex boundaries compared to the full MA mesh movement approach. Please see the full video of this case in the supplementary materials.

## 5.3 Tangled results of the UM2N model

There are in fact still cases where UM2N might fail. To investigate these, we perform additional experiments as shown in Figure 7. Starting from a rectangle, we gradually distort the geometry to a more and more non-convex shape, *i.e.*, the case becomes more challenging from top to bottom. The UM2N finally fails in the extreme case at sudden jumps in the boundary accompanied by a large variation in the required resolution as shown in the bottom row of Figure R1: a sudden constriction of the flow leads to tangling with UM2N in the two left corners of the channel. In less extreme cases, UM2N does produce a valid mesh, whereas the original MA method still fails (as seen in the second to bottom rows in Figure 7).

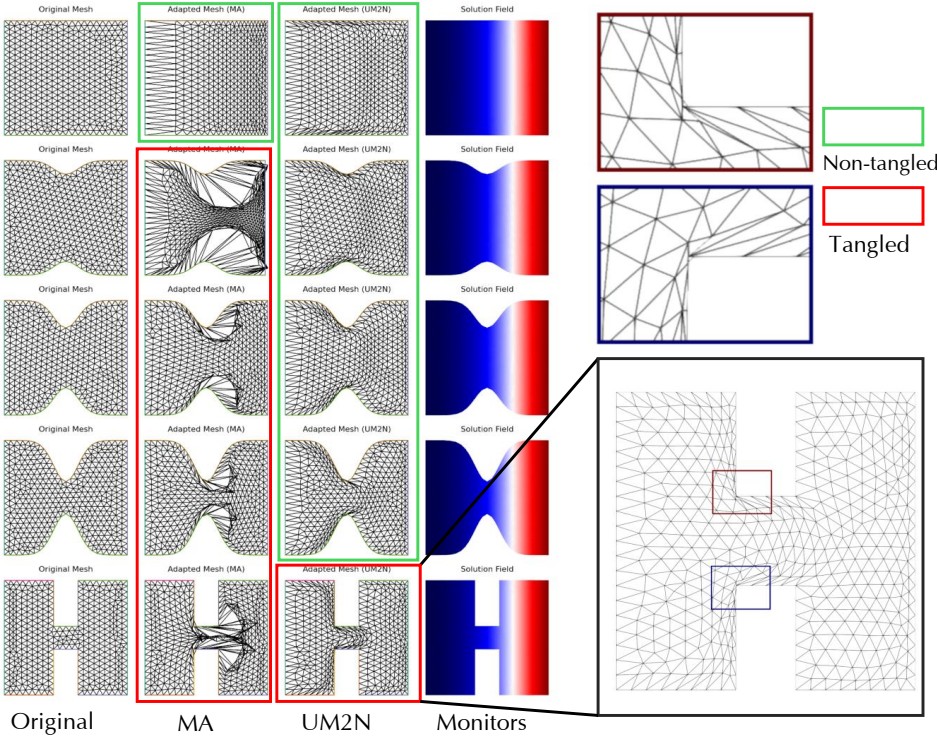

Figure 7: Tangled Case of UM2N and MA. Red/blue indicate high/low monitor values, *i.e.*, the mesh is expected to stretch towards the right direction. The case is more and more challenging from top to bottom.

## 5.4 Ablation study

**Volume loss vs coordinate loss.** We denote the UM2N variant trained with coordinate loss [19] as UM2N-coord. As shown in Table 2, UM2N-coord achieves comparable error reduction (31.21%) to the proposed UM2N (35.93%) on the Swirl case. However, it fails (*i.e.*, encounters mesh tangling) on the Cylinder case which has a more complex boundary geometry. This indicates that the element volume loss is mesh-tangling aware, *i.e.*, helps to prevent producing inverted elements.

Table 2: Ablation Study for design choices of learning monitors and element volume loss.

| Method | Helmholtz ER (%) ↑ | Swirl ER ↑ (%) | Cylinder ER (%) ↑ |
|---|---|---|---|
| UM2N-coord | 16.64 | 31.21 | Fail |
| UM2N-sol | 13.52 | -0.35 | 39.94 |
| UM2N (Ours) | **17.08** | **35.93** | **61.29** |

**Monitors vs PDE solutions as inputs.** We denote the UM2N variant using the PDE solution as input, instead of monitor function values, as UM2N-sol. The UM2N-sol is trained on solutions of the Helmholtz PDE. The UM2N-sol shows a much inferior performance (−0.35%) compared to UM2N (35.93%) on the Swirl case as well as on the Cylinder case. This indicates that learning mesh movement from monitors directly improves the generalisability of the model.

## 6   Conclusions

We introduce the Universal Mesh Movement Network (UM2N), which once trained on our generated PDE-independent dataset, can be applied in a non-intrusive, zero-shot manner to move meshes with different sizes and structures, for solvers applicable to different PDE types and boundary geometries. Our method demonstrates superior mesh movement results on Advection and Navier-Stokes PDE examples as well as a real-world tsunami simulation case. Our UM2N outperforms the existing learning based method and achieves comparable PDE error reduction to the far more costly Monge-Ampére (MA) PDE based approach. Our UM2N demonstrates effectiveness in cases with complex boundary geometries where existing learning and conventional MA based methods fail. Our method also shows significant acceleration compared to MA based mesh adaptation. See Appx. G for discussions on limitations and broader impacts.

## Acknowledgment

Thanks to Jinghan Jia for testing UM2N performance on Nvidia A100.

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

# Appendix

We include additional experimental details and results in the appendix. Detailed information on the software used is provided in Appendix A. A description of the dataset generation process is presented in Appendix B. We provide the details of experimental settings of cases in the paper in Appendix C. To provide a more holistic analysis for the proposed UM2N, we include evaluations across various test cases, performance benchmarks, more ablation studies, and assessments on low-quality initial meshes in Appendix D. To demonstrate the the universal ability of the UM2N, additional experiments conducted under diverse settings are detailed in Appendix E. We introduce the mesh movement-enhanced simulation, illustrating the non-intrusive integration of UM2N into a numerical simulation pipeline in Appendix F. A discussion about the limitations and broader impacts of our approach is provided in Appendix G.

## A    Software used in the paper.

In this paper, PDEs are solved using finite element methods with *Firedrake* [44]. Firedrake is written in Python, uses the *Unified Form Language (UFL)* [45] domain-specific language to represent finite element forms, and automatically generates C kernels for computational efficiency. Firedrake uses *PETSc* [46] for solving linear and nonlinear equations that result from discretisation, as well as for its underlying unstructured mesh representation.

Original meshes (*i.e.*, meshes before adaptation) are generated by either Firedrake or Gmsh [47]. Gmsh is a meshing software especially designed for finite element methods, providing access to several meshing algorithms. In the case of the Tōhoku tsunami simulation, the mesh of the Japan Sea is generated using *qmesh* [48] – a meshing tool specifically designed for coastal ocean modelling applications, which manages GIS (Geoscientific Information System)-based datasets to construct a valid input geometry used by Gmsh to generate the mesh.

The conventional mesh movement strategy used to generate training data is implemented in *Movement* [40], which is itself written in UFL and Firedrake. Movement implements several mesh movement approaches, including two based on solutions of Monge-Ampère type equations. In this paper, we use the 'relaxation' approach, which solves the nonlinear MA equation in an iterative fashion by introducing a pseudo-timestep (see [36] for details).

The code used to produce the results presented in this paper is publicly available at `https://github.com/mesh-adaptation/UM2N`. (Release [49].) Code documentation is hosted at `https://mesh-adaptation.github.io`.

## B    Dataset description.

### B.1    Dataset generation pipeline.

We developed a pipeline for generating PDE-independent training data and further research. Through our pipeline, generic fields are generated and can be utilized as solutions of any PDE. The monitor values computed from the generic field are used to guide the MA-based method to perform mesh adaption on the original mesh.

We stack $N$ randomly generated Gaussian distributions to form the generic field $u$. Concretely, they can be generated by the formula:

$$u = \sum_{k=1}^{N} \exp\left(\frac{(x - \mu_x)^2}{\sigma_x{}^2} + \frac{(y - \mu_y)^2}{\sigma_y{}^2}\right), \tag{A1}$$

where $\mu_x$ and $\mu_y$ denote the means of the distribution along two orthogonal directions in a 2D domain, while $\sigma_x$ and $\sigma_y$ represent the respective standard deviations.

All training data are generated within a 2-D unit square domain. Equivalent size $n \times n$ meshes ($n \in \{18, 20\}$) are generated for model training. Here we show samples of the generated fields in Fig. A1.

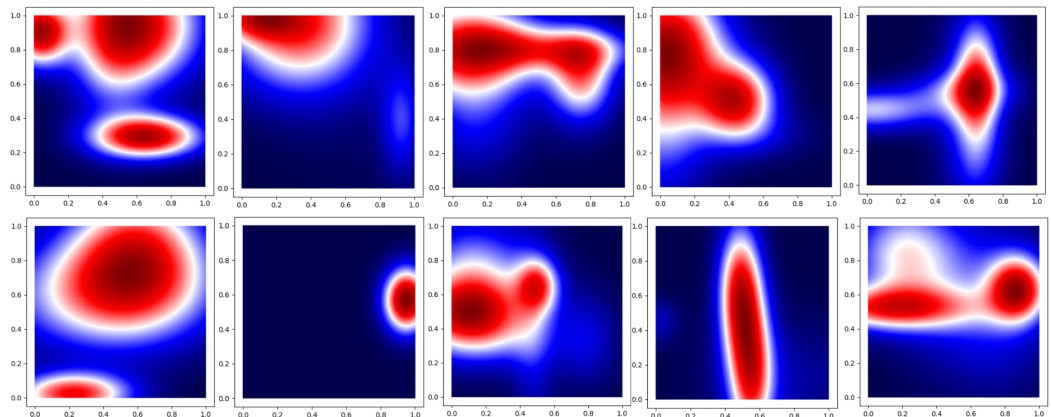

Figure A1: Samples of generated generic fields.

## B.2 Evaluation data for the Tōhoku tsunami simulation.

As mentioned in Sec. A, the mesh of the Japan Sea is generated using qmesh [48]. For this, we provide qmesh with coastline data from [50]. Bathymetry (sea bed topography) data is taken from the ETOPO1 dataset [51]. A highly idealised tsunami source condition is used, comprised of a sum of Okada functions [52].

## C  Details of experimental settings.

### C.1  Helmholtz.

The Helmholtz equation used in the paper is defined as:

$$-\nabla^2 u + u = f, \quad \nabla u \cdot \overrightarrow{n} = 0 \text{ on } \Gamma, \tag{A2}$$

where $f$ is the source term and $\Gamma$ denotes the boundary.

### C.2  Swirl.

We use a similar setting to the unsteady 2D linear advection case in [18].

**Initial Condition.**

$$u_0 = \exp\left(-\frac{1}{2\sigma^2}(\sqrt{(x-x_0)^2 + (y-y_0)^2} - r_0)^2\right). \tag{A3}$$

The initial condition defines a ring centered at $(x_0, y_0)$ with an inner radius $r_0$. The thickness of the ring is $3\sigma$. In this paper, $r_0 = 0.2$, $(x_0, y_0) = (0.30, 0.30)$, and $\sigma = 0.05/3$.

**Velocity Field.**

$$\boldsymbol{c}(x, y, t) = \left(\frac{9}{10}a(t)\sin^2(\pi x)\sin(2\pi y), -\frac{9}{10}a(t)\sin^2(\pi y)\sin(2\pi x)\right). \tag{A4}$$

In this paper, $0 < t < 1$, if $t < 0.5$ then $a(t) = 1$, if $0.5 <= t < 1$, $a(t) = -1$, *i.e.*, the velocity direction is reversed at $t = 0.5$

**Monitor Function.**

$$m(\mathbf{x}) = 1 + \max\left(\alpha\frac{\|H(u)\|}{\max\|H(u)\|}, \beta\frac{\|G(u)\|}{\max\|G(u)\|}\right), \tag{A5}$$

where we set $\alpha = 5$ and $\beta = 10$. The $\|H(u)\|$ and $\|G(u)\|$ are Hessian norm and gradient norm respectively.

Table A2: Performance benchmark on test case shown in Figure. A3.

| Hardware | MA CPU | UM2N CPU | UM2N GPU (Nvida RTX 3090) | UM2N GPU (Nvida A100) |
|---|---|---|---|---|
| Avg. inference time (s) | 114.3 | 1.113 | 0.0487 | 0.0342 |
| Speedup | - | $\sim 102\times$ | $\sim 2347\times$ | $\sim 3342\times$ |

## C.3 Flow past a cylinder.

**Setting** The Reynolds number is 100, a quadratic velocity profile is imposed at the left boundary as inflow condition. At the right boundary a zero pressure outflow condition is applied. The cylinder and top-bottom surface are imposed with non-slip boundaries.

## D   More Evaluations.

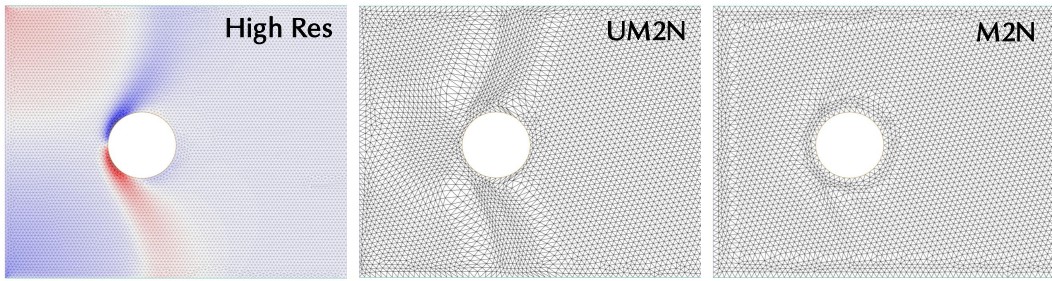

Figure A2: Non-tangled results of M2N on flow-past-a-cylinder case.

## D.1   Non-tangled results of other methods on the flow-past-cylinder.

We additionally perform a non-tangled flow-past-cylinder shown in Fig. A2. We simplified the setting: lower the Rrenynolds number by reducing the inflow velocity and set the top and bottom slip boundary, which finally makes it a laminar flow. As shown in Fig A2, both the UM2N and M2N give non-tangled meshes and it can be observed that UM2N better captures the dynamics *i.e.*, adapts to the PDE solution of stable state compared to the High

Table A1: Measure between UM2N and M2N model output mesh on the flow-past-a-cylinder in Fig. A2.

| Property | UM2N ER $\uparrow$ (%) | M2N ER $\uparrow$ (%) |
|---|---|---|
| Vorticity | **27.43** % | 10.22 % |

Res solution. The quantitative results shown in the Table A1 also indicate that UM2N perform better than M2N regarding to error reduction.

## D.2   Performance benchmark.

To evaluate the runtime performance of UM2N, we constructed a benchmark case based on the Swirl scenario, consisting of 11,833 vertices and 23,668 triangular elements. We extended the one-ring setup to a nine-ring configuration. In this benchmark, we compare the UM2N output mesh and the reference mesh generated by the MA method, as well as the solutions obtained on these meshes, against those computed on high-resolution and original meshes as shown in Fig. A3.

We tested the inference time of UM2N on two GPUs (Nvidia GeForce RTX 3090 and Nvidia A100) and compared its performance to the MA method. For a fair comparison, we also included CPU benchmarks, as the current state-of-the-art MA implementation is limited to CPU execution. The CPU used for testing was an 11th Gen Intel(R) Core(TM) i9-11900K @ 3.50GHz. The performance results are presented in Table A2.

The results indicate that UM2N achieved more than 2000x speedup on the Nvidia RTX 3090 and over 3000x on the Nvidia A100, compared to the MA method. Even on the CPU, UM2N demonstrated an approximate 100x speedup. In summary, UM2N provides comparable solution accuracy while delivering a significant performance improvement.

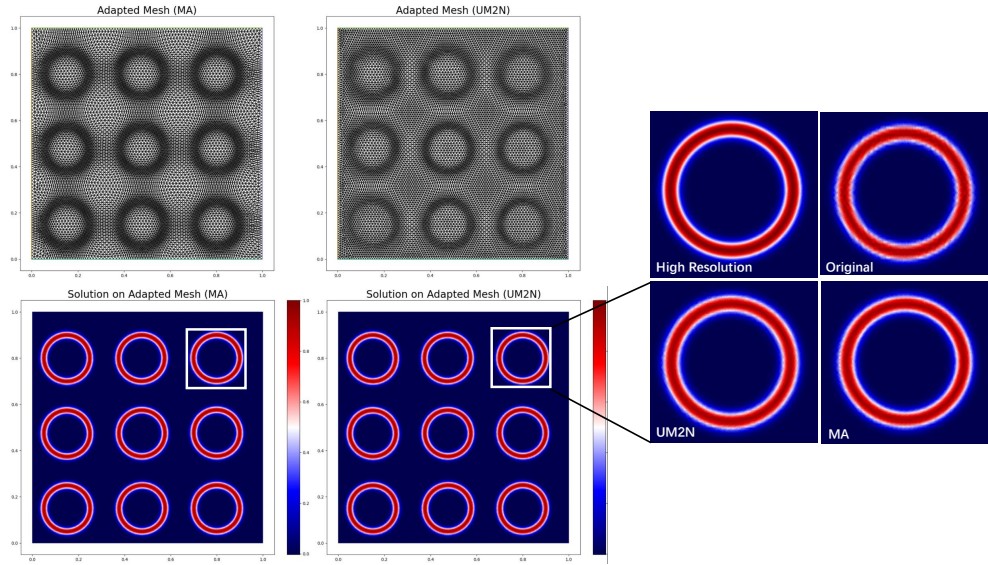

Figure A3: Visualization of UM2N output mesh and reference mesh from MA. Comparison between solutions obtained on high resolution, original, UM2N and MA meshes.

## D.3 Boundary layer case.

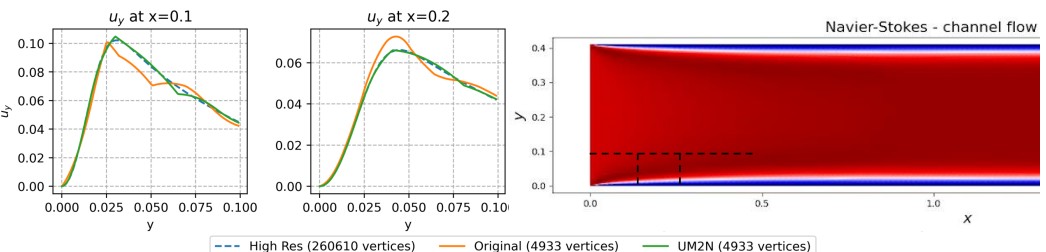

Figure A4: Boundary Layer. The two plots at left visualize the $u_y$ from boundary to $y = 0.1$. It can be observed that UM2N improves the accuracy of the velocity obtained on uniform mesh compared to High Res solution.

In our flow past the cylinder case, at the top and bottom we imposed non-slip boundary conditions which already promotes a boundary layer similar to the suggested flow over a flat plate. For a clearer illustration, we have now additionally conducted a flow past two parallel plates experiment. We observe that the velocity parallel to the boundary ($u_y$) obtained from UM2N moved mesh better aligns with the high resolution results compared to the results on uniform mesh as shown in the Fig. A4.

## D.4 Poor quality initial mesh.

We include an additional experiment starting from a poor quality initial mesh as shown in Fig. A5. The input mesh has highly anisotropic elements and several vertices with valence 6. This can certainly be considered a "low quality" mesh. Consider the monitor function to be the $L^2$-norm of the recovered gradient of the solution of an anisotropic Helmholtz problem. With this monitor function, we tried applying conventional MA solvers and found that both the quasi-Newton and relaxation approaches failed to converge and/or resulted in tangled meshes, even though the elements are already aligned with the anisotropy of the Helmholtz solution. The UM2N approach, however, was able to successfully apply mesh movement without tangling. We would also like to clarify that our work targeting mesh movement that dynamically adapts to a PDE solution, which is not a replacement for mesh generation in general, so we would always expect a reasonable input mesh when applying our model.

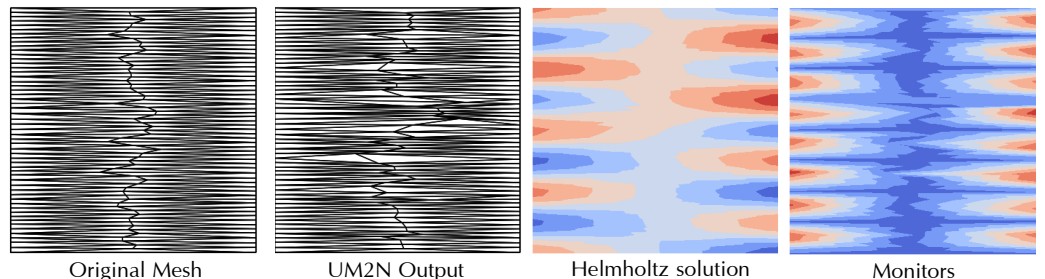

|  | Original Mesh | UM2N Output | Helmholtz solution | Monitors |

Figure A5: A very poor initial mesh case. UM2N perform mesh movement for an anisotropic Helmholtz problem.

| Method | Swirl ER ↑ (%) |
|---|---|
| UM2N-w/o-Decoder | 10.68 |
| UM2N-w/o-GT | 12.35 |
| **UM2N (Ours)** | 35.93 |
| MA (reference) | 37.43 |

(a) Error reduction of UM2N variants on Swirl case.

(b) Moved from UM2N variants and reference mesh from MA on Swirl case.

Figure A6: Ablation of network components. Full UM2N gives the best error reduction.

## D.5 Further Ablation studies.

We perform an additional ablation study of our network architecture on the Swirl case. We construct two models, which replace the GAT Decoder with one layer GAT denoted as UM2N-w/o-Decoder and remove the graph transformer denoted as UM2N-w/o-GT. The results are shown in the Table A6a.It can be observed that both variants give worse results compared to the full UM2N. A visualization is also shown in Fig. A6b. It can be observed that without the Decoder, the model distorts the shape of the ring, i.e., missing relative information between vertices; without the graph transformer, the model fails to capture the details of the ring shape.

## E Additional experimental results.

**Flow past multiple cylinders.** The Fig. A7 shows a challenge scenario with 5 cylinders in the domain. The proposed UM2N can move mesh capturing the complex dynamics without mesh tangling.

**Flow past V-formation cylinders.** The Fig. A8 shows a bio-inspired V-formation setting. The proposed UM2N can move mesh capturing the wake flows.

**Flow past cylinders and squares.** The Fig. A9 shows the proposed UM2N can handle another challenging scenario with a mixture of cylinders and squares of different sizes.

**Lift coefficient ($C_L$) of flow past cylinder.** In Fig. A10, we visualized the lift coefficient over timestep across different methods.

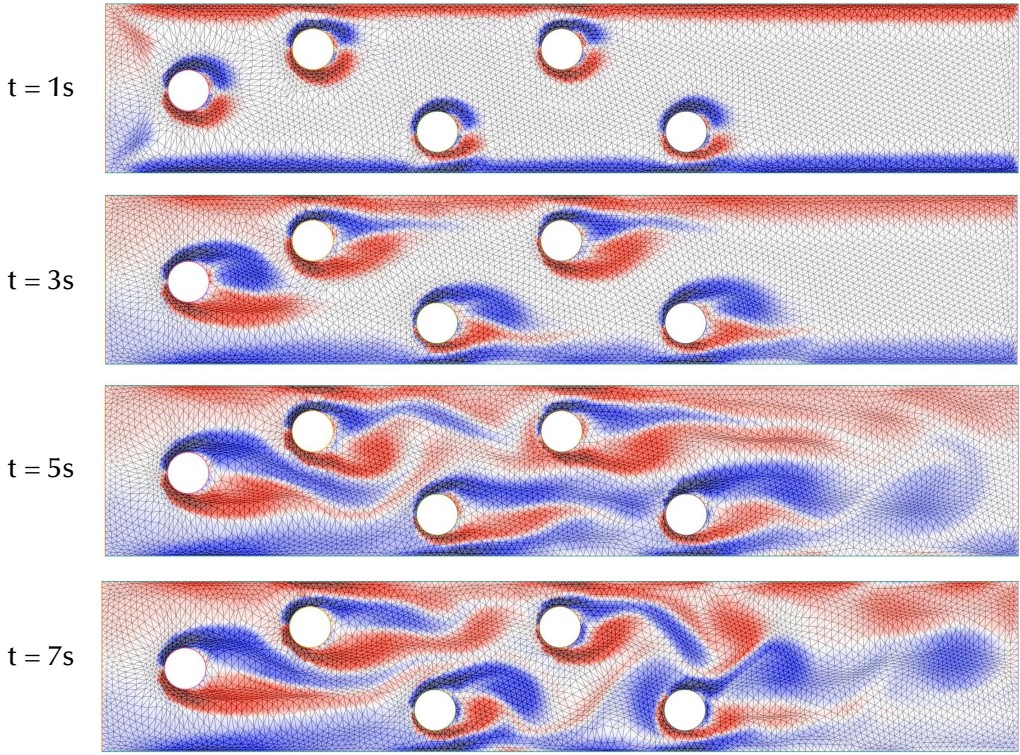

Figure A7: Multiple Cylinders

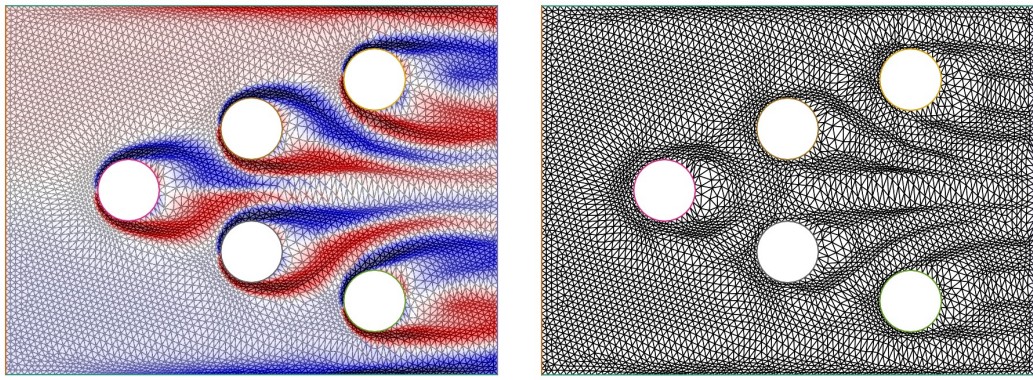

Figure A8: V-formation

# F    Mesh movement enhanced simulation.

The Mesh Movement Enhanced Simulation Algorithm, as outlined in Alg. 1, integrates our mesh adaption strategies into the simulation process to enhance the quality and accuracy of computational solutions. The proposed MMES procedure can be adapted into almost any time-dependent numerical simulation processes non-intrusively.

For a given time-dependent PDE $\mathcal{P}$ and an initial mesh $\boldsymbol{\xi}_{\text{init}}$, the algorithm evaluates the monitor function $m$ based on the solution field of the PDE at each time step $t$. The monitor value $m$ is then used by our proposed model to guide mesh movement. For robustness, a mesh integrity check is performed before time stepping (a tangled mesh will interrupt the procedure).

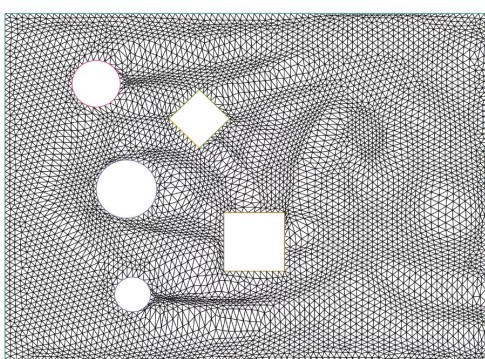

Figure A9: Cylinder and Square

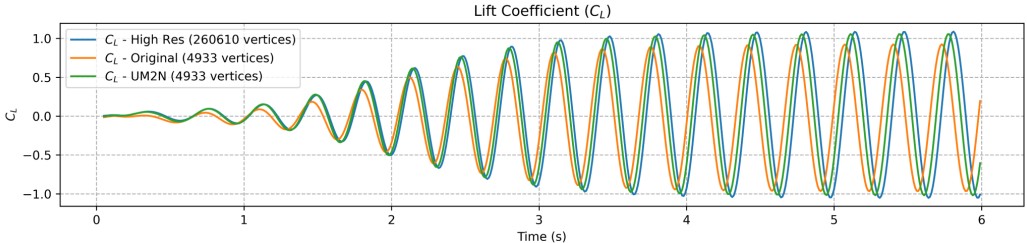

Figure A10: Results of $C_L$ coefficients. The blue, orange and green lines show the drag $C_L$ coefficients obtained on a high resolution fixed mesh, UM2N adapted mesh and the original coarse mesh for the first 6 seconds (i.e., 6000 steps). It can be observed that the UM2N output mesh improves the accuracy of $C_L$ in magnitude and periodicity compared to the original mesh.

## G   Limitations and broader impacts.

**Limitations.** 1) The GAT to an extent constrains the mesh movement to lie in a restricted range, which may be less effective in scenarios requiring substantial large deformation within a single iteration of the mesh mover. 2) There is no theoretical guarantee that mesh tangling will be completely prevented. 3) The model's best performance relies on a manually chosen monitor function, though a gradient based monitor function can yield reasonable results. 3) Given a low-quality or extremely coarse original mesh, the efficacy of the proposed method may be constrained by the inherent

---

**Algorithm 1** Mesh Movement Enhanced Simulation

1: **procedure** SIMULATION($\mathcal{P}, \boldsymbol{\xi}_{\text{init}}, T$)
2:     $\boldsymbol{\xi}_t \leftarrow \boldsymbol{\xi}_{\text{init}}$
3:     **while** $t < T$ **do**
4:         $u \leftarrow$ PDESolver($\mathcal{P}, \boldsymbol{\xi}_t$)
5:         $m_{\boldsymbol{\xi}} \leftarrow$ MonitorFunc($u, \boldsymbol{\xi}_t$)
6:         $\boldsymbol{\xi}_{t+1} \leftarrow$ UM2N($\boldsymbol{\xi}_t, m_{\boldsymbol{\xi}}$)
7:         MeshIsValid $\leftarrow$ CheckMeshNotTangle($\boldsymbol{\xi}_{t+1}$)
8:         **if** MeshIsValid **then**
9:             $\boldsymbol{\xi}_t \leftarrow \boldsymbol{\xi}_{t+1}$
10:         **else**
11:             **Break**
12:         **end if**
13:         TimeStepUpdate()
14:     **end while**
15: **end procedure**

---

limitation on the degree of freedom count, as it redistributes rather than increases the number of mesh vertices. A promising future direction could be exploring the integration of mesh movement with $h$-adaptation, potentially developing a hybrid $hr$-adaptation approach that combines the advantages of both techniques.

**Broader Impacts.** This research contributes to the field of AI for physical simulation by introducing an advanced learning-based mesh movement method that enhances the accuracy and efficiency of numerical simulations. The primary application of this method lies in areas requiring high-fidelity simulations of physical phenomena, such as geophysics, renewable energy, climate modeling, and aerodynamics. This method can aid in better weather or hazards forecasting, more efficient energy harvesting, fast prototyping of aircraft design etc.

