# OpenReview forum: "Towards Universal Mesh Movement Networks"
_NeurIPS.cc/2024/Conference — NeurIPS 2024 spotlight_

### Official Review · Reviewer_Jzyt · 2024-06-21

**Soundness:** 3
**Presentation:** 3
**Contribution:** 3
**Rating:** 6
**Confidence:** 3

**Summary:**

The authors propose a method for general mesh movement. They design a network that takes the original mesh and the monitor values as input to predict the mapping for the adapted mesh. This makes it able to handle any mesh without the need of training PDE-type-specific models.

**Strengths:**

1. The proposed method can handle mesh with different geometries and PDE with different type using a single model. The model only needs to be trained once.
2. Another advantage is its time cost is much lower than the traditional non-learned Monge-Ampère without the sacrifice of the movement accuracy.
3. Compared with prior works, ie, MA and M2N, it is more robust and easier to converge.
4. The paper is well written and easy to follow.

**Weaknesses:**

The idea of this paper is interesting to me, and the results, especially those shown in the supplementary video, are pretty nice. I believe the proposed method can have many potential application in mesh formation and the modeling of physical phenomena. I didn't see any obvious weaknesses from this paper. However, I have to admit that I'm not the expert in this field.

**Questions:**

1. The network takes the original mesh and the monitor values $m$ as input. $m$ is the values for the original mesh or the target mesh? If it is the value of the original mesh, then how does the network know the target? If it is the value of the target mesh, then the symbol of $m_\xi$ is a bit misleading, since $\xi$ denotes the original mesh.
2. When using chamfer distance for the optimization of mesh matching, the result is easy to be stuck at the local minimum. Is it the case for the training of the proposed method?

**Limitations:**

In the section of Limitation, the authors mentioned that a low-quality original mesh can lead to poor prediction. A visualization of it will be nice.

---

> ### Author Rebuttal · Authors · 2024-08-07
>
> We greatly appreciate your insightful comments that precisely recognize the strengths of our work!  We are also very happy the Reviewer Jzyt checked our supplementary video and found it nice and useful. Below, we provide our point-to-point responses to the comments, denoted by **[W]** for weaknesses and **[Q]** for questions.
>
> **Response to Q1**:
> The monitor values 𝑚 correspond to the original mesh. Given a time-dependent PDE, the monitor values $𝑚$ at the first time step are computed based on the initial condition on the original mesh. Then the model takes in the monitor values $𝑚$ as input (with other features) and outputs a moved mesh. The PDE is solved based on this moved mesh and the solution obtained will be used for computing monitor values for next time-step.
>
> The target meshes only come in during the training process: the moved mesh is compared to the target mesh with element volume loss and Chamfer distance. The networks get to know the target during the training. In other words, the monitor values are part of input features and are defined in the nodes of the input  meshes and not those of the target meshes. This indeed explains the notation $m_ξ$ with the monitor values being associated with the input mesh. The training process can be explained to teach the model to learn to solve an optimal transport problem according to equidistribution theory given the monitor values and other related features.
>
> **Response to Q2**:
> Thanks for raising this question. We acknowledge that Chamfer distance-based optimization for mesh matching or point cloud registration can potentially lead to convergence at local minima. However, in our case, we did not observe significant issues of being trapped in poor local minima during the training process, though we cannot guarantee that a global minimum was reached.
> We believe there are two primary reasons for this. First, the initial alignment between the original uniform meshes and the target meshes in our training dataset is not significantly different. This likely increases the chances of finding well-matched vertices when searching for nearest points using Chamfer distance. Second, in addition to the Chamfer distance, we employed an element volume loss, which provides additional information beneficial for vertex matching.
>
> **Response to limitation**:
> Thanks for the suggestions. We include an additional experiment about poor quality initial mesh as shown in Figure R4.
> The input mesh has highly anisotropic elements and several vertices with valence 6. This can certainly be considered a “low quality” mesh. Consider the monitor function to be the l2-norm of the recovered gradient of the solution of an anisotropic Helmholtz problem. With this monitor function, we tried applying conventional MA solvers and found that both the quasi-Newton and relaxation approaches failed to converge and/or resulted in tangled meshes, even though the elements are already aligned with the anisotropy of the Helmholtz solution. The UM2N approach, however, was able to successfully apply mesh movement without tangling.
> We would also like to clarify that our work targeting mesh movement that dynamically adapts to a PDE solution, which is not a replacement for mesh generation in general, so we would always expect a reasonable input mesh when applying our model.

---

> > ### Comment · Reviewer_Jzyt · 2024-08-09
> >
> > Thanks for your clarification for my questions. Please include the new results in the revised paper.

---

> > > ### Author Response · Authors · 2024-08-09
> > > **Thank you!**
> > >
> > > Dear Jzyt,
> > >
> > > We sincerely thank you for your insightful comments with us! We commit to incorporate the new results in the revised paper.
> > >
> > > Best regards,
> > > Authors

---

### Official Review · Reviewer_jEN5 · 2024-07-12

**Soundness:** 3
**Presentation:** 3
**Contribution:** 3
**Rating:** 6
**Confidence:** 2

**Summary:**

This paper introduces a learnable model for mesh movement - which is a method for improving efficiency of a PDE solver by moving mesh nodes, while keeping the topology fixed. The proposed model itself is a two-stage neural architecture comprising a transformer encoder - taking a representation of the input mesh - and a graph attention network as a decoder outputting an updated mesh. The resulting model is independent of specific PDE type and thus can be applied to novel problem types in zero-shot manner. Quantitative evaluation is conducted on multiple synthetic and one realistic (tsunami prediction) tasks.

**Strengths:**

- Proposed method provides a way to improve PDE solvers agnostic to the underlying PDE type. A zero-shot model like this could be extremely useful in a lot of engineering fields.
- Quantitative results indicate that the method is outperforming existing method for mesh movement and leads to improved surrogate modeling.

**Weaknesses:**

- Although overall the architecture makes sense, there is no ablation study conducted on the individual parts of the architecture.
- Similarly, there are no baselines to off-the-shelf graph convolutional or transformer methods (any recent method e.g. for semantic mesh segmentation would work here?).
- The model is only concerned with vertex movement of the model, which means there is no guarantee that there won't be any degraded triangles, which can influence the quality of the dowstream solutions. This could be a fundamental flaw of purely data-driven methods since there is not simple way to enforce constraints at test time.
- The scale of the dataset seems to be very small for the model to be directly applicable in realistic scenarios - both in terms of number of samples and number of vertices.
- Model relies on a transformer encoder, which will not scale to meshes with large number of vertices.

**Questions:**

- In figure 6, you demonstrate that um2n is better at handling complex boundaries. From what I can see there are no guarantess for your purely data-driven method to produce non-tangled meshes? Do you have an intuition to why this is the case?
- Would this model work for larger scale problems with >1M?

**Limitations:**

Limitations are discussed.

---

> ### Author Rebuttal · Authors · 2024-08-07
>
> We thank Reviewer jEN5 for acknowledging the contributions, soundness, and presentation quality of our paper. And greatly appreciate Reviewer jEN5 for proposing these insightful questions. Below, we provide our point-to-point responses to the comments, denoted by **[W]** for weaknesses and **[Q]** for questions.
>
>
> **Response to W1**: We perform an additional ablation study of our network architecture on the Swirl case. We construct two models, which replace the GAT Decoder with one layer GAT denoted as UM2N-w/o-Decoder and remove the graph transformer denoted as UM2N-w/o-GT. The results are shown in the Table R1.It can be observed that both variants give worse results compared to the full UM2N. A visualization is also shown in Figure R2. It can be observed that without the Decoder, the model distorts the shape of the ring, i.e., missing relative information between vertices; without the graph transformer, the model fails to capture the details of the ring shape.
>
>
> **Response to W2**: To the best of our knowledge, there are not a lot of works addressing the end-to-end learning-based r-adaptation problem. Many existing works focus on end-to-end surrogate neural PDE solvers. The models proposed in these works try to predict the PDE solutions directly, therefore it is hard to make a fair comparison as our UM2N predicts the mesh i.e., PDE discretization. To compare with them qualitatively, these methods have advantages in efficiency. On the other hand, our approach has the advantage that PDE solutions obtained are physically plausible based on guarantees from classical numerical analysis regarding the PDE discretization, whereas such guarantees are absent or enforced in a weaker sense in directly learned PDE solutions.
>
> For the network architecture, we apply the graph transformer as encoder for its powerful expressivity, and apply the graph attention network as decoder because it can naturally help alleviate mesh tangling. There are existing works such as efficient transformers from natural language processing or computer vision community, which provide promising future directions to improve the efficiency of our model. The semantic mesh segmentation mentioned by the Reviewer jEN5, although not directly applicable to our problem, would also be an interesting direction to explore.
>
> **Response to W3/Q1**: We agree that there are no guarantees for our method to produce non-tangled meshes as we have also noted in the limitations section of our paper. In Figure 6, we claim that UM2N is better at handling complex boundaries compared to the MA method. The MA method requires smooth boundaries and convex domains. Its behaviour can be observed from our additional experiments shown in Figure R1. It produces good results for the rectangle domain (shown in the first row) but outputs a highly tangled mesh given a non-convex domain (shown in the following rows).
>
> The intuitions behind why our data-driven UM2N can handle complex boundaries can be divided into two aspects. The first is that we use Graph Attention Network (GAT) to build the mesh deformer. Therefore, the coordinate of each vertex is updated by the weighted sum of its neighbors, the weights (or coefficients) of which are determined by the GAT module. This guarantees each vertex to only move within the convex hull of its neighboring vertices, hence effectively alleviating mesh tangling issues. In addition, we utilize element volume loss penalizing negative element volumes (i.e., inverted elements), which helps reduce mesh tangling.
>
>
> **Response to W4/W5/Q2**:
> As for the scale of the dataset, we consider the test case shown in the paper -  Cylinder (~5k vertices, ~10k triangles), Tsunami ( ~8k vertices, ~16k triangles) - to have moderate degrees of freedom. The tsunami case is already a real-world scenario application in the ocean modelling community. To show a case with more degree of freedom, we further apply our trained model on a flow past cylinder case with ~11k vertices and ~22k triangles as shown in Figure R5. It can be shown that our UM2N works well in this case.
>
> As for the large case such as >1 million vertices/triangles mentioned by the reviewer, the model can be naturally applied to larger scale problems as there is no limitation for the input length of graph transformer and graph neural network but the GPU memory is the limit. We performed stress tests using a time-independent Helmholtz case on our RTX 3090 24 GB GPU and observe that the maximum scale of the problem that can be run on this GPU has ~50k number of elements. The inference time is ~760 ms, the MA method on the same problem requires ~37800 ms with a residual threshold 1e-4. Therefore, given our current limited computational resource, it is hard for us to construct a case with 1 > million vertices. Considering this limitation, applying the memory efficient transformer (linear-attention transformer for further improvement of the inference efficiency) is targeted as future work.

---

> > ### Comment · Reviewer_jEN5 · 2024-08-12
> >
> > Thanks for the detailed response. This addresses my questions/concerns so I am keeping my original score.

---

> > > ### Author Response · Authors · 2024-08-13
> > > **Thank you**
> > >
> > > Thanks for the reviewer's acknowledgment of our efforts to address the concerns!

---

### Official Review · Reviewer_yaVX · 2024-07-22

**Soundness:** 4
**Presentation:** 3
**Contribution:** 3
**Rating:** 6
**Confidence:** 3

**Summary:**

The paper introduces the Universal Mesh Movement Network (UM2N), a deep-learning model that enhances solving Partial Differential Equations (PDEs) through adaptive mesh movement. UM2N, with a Graph Transformer encoder and a Graph Attention Network (GAT) based decoder, is trained on a PDE-independent dataset, allowing zero-shot application to various PDEs and geometries without retraining. The model uses element volume loss to improve robustness and reduce mesh tangling.

**Strengths:**

1/ Originality:

The paper presents a novel approach that advances the field of adaptive mesh movement for solving Partial Differential Equations (PDEs). By combining a Graph Transformer encoder with a Graph Attention Network (GAT) decoder, the authors develop a unique method that can be applied zero-shot to various PDE types and boundary geometries without retraining. This approach overcomes the limitations of traditional and existing learning-based methods, which often require costly retraining and struggle with complex geometries.

2/ Quality:

The quality of the paper is good, demonstrated through comprehensive evaluations on different PDE types, including advection, Navier-Stokes, and a real-world tsunami simulation. The experiments clearly show that UM2N outperforms existing methods in terms of accuracy, efficiency, and robustness. The use of a PDE-independent dataset for training and the adoption of element volume loss for reducing mesh tangling further highlight the thoroughness and rigor of the approach.

3/ Clarity:

The methodology is clearly explained, and the results are presented in a way that effectively supports the claims made.

4/ Significance:

The significance of this work is substantial, as it addresses a fundamental challenge in scientific and engineering simulations:
solving PDEs efficiently and accurately. By enabling zero-shot application to various PDEs and geometries, UM2N offers a versatile tool that can be integrated into a wide range of numerical solvers, potentially benefiting numerous applications in geophysics, renewable energy, climate modeling, and aerodynamics. The improvements in mesh movement accuracy and reduction in computational costs have broad implications for advancing simulation technologies and their applications.

5/ Learning-based mesh generation and adaptation:

One of the primary advancements is the use of a PDE-independent dataset for training. This allows UM2N to generalize across various PDE types and boundary geometries without the need for retraining. This is a significant improvement over existing methods that typically require retraining for new scenarios, making UM2N more versatile and practical for real-world applications. Another key strength is the incorporation of element volume loss in the training process. Unlike coordinate loss used in previous methods, element volume loss not only supervises the mesh movement but also penalizes negative volumes. This effectively reduces mesh tangling and enhances robustness.

The architectural choice of combining a Graph Transformer encoder with a Graph Attention Network (GAT) decoder is particularly well-suited for handling the irregular structures of unstructured meshes. This graph-based architecture allows UM2N to capture both local and global information more effectively, leading to more accurate and efficient mesh movements. The attention mechanisms within these networks help in managing the complex dependencies and interactions within the mesh elements.

Moreover, UM2N's ability to perform zero-shot generalization is another significant strength. The model can be applied to new, unseen problems without the need for additional training. This capability is particularly beneficial in practical applications where retraining models is computationally expensive and time-consuming.

By using a learning-based approach for mesh movement, UM2N significantly improves the accuracy of numerical solutions while also reducing computational costs. This balance between accuracy and efficiency is crucial for advancing simulation technologies in various scientific and engineering domains. The method's ability to enhance accuracy without incurring additional computational overhead makes it an important tool for high-fidelity simulations.


6/ Monge-Ampère PDE-solver point of view:

From the perspective of Monge-Ampère, UM2N enhances practical applicability by addressing computational and robustness challenges through a novel learning-based approach. Traditionally, Monge-Ampère-based methods offer attractive theoretical properties such as equidistribution and optimal transport but are computationally expensive and struggle with complex geometries. UM2N mitigates these limitations by decoupling the underlying PDE solve from the mesh movement process, focusing on learning the auxiliary Monge-Ampère equation.

The integration of a Graph Transformer and GAT-based architecture allows UM2N to handle the complexities of mesh movement efficiently. The Graph Transformer captures both local and global information from the mesh, which is essential for accurately solving the Monge-Ampère equation. It processes the mesh as a graph, taking into account the positional relationships and interactions between mesh elements. The GAT-based decoder then uses this rich feature set to move the mesh nodes in a way that aligns with the desired mesh density distribution.

UM2N's use of the element volume loss function is a significant technical advancement. This loss function is inspired by the equidistribution principle of the Monge-Ampère method, which aims to distribute the monitor function values uniformly across the mesh. By focusing on the volumes of the mesh elements rather than their coordinates, the element volume loss function ensures that the mesh adapts according to the desired density distribution. This approach also penalizes negative volumes, which helps prevent mesh tangling a common issue in traditional Monge-Ampère-based methods.

Another important detail is UM2N's training on a PDE-independent dataset, which allows it to generalize well to different PDEs and boundary conditions without retraining. This is a significant improvement over traditional Monge-Ampère methods, which often require problem-specific adjustments. UM2N's ability to perform zero-shot generalization makes it highly versatile and practical for a wide range of applications.

UM2N's robustness is further demonstrated through its successful application to scenarios with complex boundary geometries, such as the Tohoku tsunami simulation. Traditional Monge-Ampère methods often fail in such scenarios due to the challenges in solving the non-linear Monge-Ampère equation under these conditions. UM2N, however, effectively manages these complexities, maintaining high-quality mesh adaptation even in challenging geometries.

**Weaknesses:**

1/ Comparison with State-of-the-art methods:

While UM2N is compared with a few existing methods, the comparison could be broadened to include more recent state-of-the-art approaches. This would provide a clearer context for UM2N's performance and highlight its relative advantages and limitations more comprehensively. Expanding the experimental section to include comparisons with more recent and relevant learning-based mesh movement methods would be valuable. If new experiments are not feasible during the rebuttal, a detailed qualitative comparison discussing potential strengths and weaknesses based on the literature can be provided.

2/ Ablation Studies:

The ablation studies are somewhat limited. A more comprehensive analysis of different components of UM2N, such as the impact of various hyperparameters, the contribution of each network component (Graph Transformer vs. GAT), and the sensitivity to different types of monitor functions, would provide deeper insights into the model's workings. The authors could add more ablation studies focusing on hyperparameter tuning, the role of different network components, and the sensitivity analysis to different monitor functions.


3/ Handling extreme geometries and mesh qualities:

The current paper does not extensively explore how UM2N handles extremely poor-quality initial meshes or highly irregular and extreme geometries. These scenarios are crucial for practical applications, as real-world problems often involve complex and challenging geometrical domains. Poor-quality initial meshes, characterized by elements with highly skewed aspect ratios, large variations in element sizes, or even degenerate elements, can significantly complicate the mesh movement process. For instance, elements with aspect ratios far from unity can lead to numerical instabilities and inaccurate solutions due to poor approximation properties. Large variations in element sizes can cause localized errors to propagate or lead to uneven error distribution across the mesh, while degenerate elements with nearly zero area or volume can invert or fold during movement, leading to invalid meshes.

Similarly, highly irregular geometries, such as those with sharp corners, narrow channels, intricate boundary shapes, or complex topological features like holes and multiple connected components, can test the robustness and adaptability of mesh movement algorithms. Sharp corners and narrow channels require fine mesh resolution to capture the geometric details accurately, while avoiding mesh element inversion or tangling. Intricate boundary shapes and complex topological features increase the difficulty of maintaining mesh quality and conformity to the domain boundaries during adaptation.

A. Aspect Ratio and skewness:

Maintaining aspect ratios close to one is crucial to minimize numerical errors. High aspect ratios can lead to poor interpolation properties, making it challenging to preserve or improve aspect ratios during significant mesh adjustments.

High skewness causes poor approximation and solver instability. Controlling skewness during mesh movement to prevent highly distorted elements is critical.

Suggestion: implement a loss term that penalizes high aspect ratios and skewness during training to encourage well-shaped elements.

B. Element size variation:

Uniformity and adaptivity:
Balancing element sizes to ensure smooth transitions between refined and coarse regions is complex. Small elements can lead to computational overhead, while large elements might miss critical solution features.

Dynamic adaptation:
Efficiently refining in regions of high gradients and coarsening elsewhere requires dynamic strategies, balancing computational cost and accuracy.

Suggestion:
Use adaptive mesh refinement algorithms that dynamically adjust element sizes based on local error estimates and monitor function gradients.

C. Mesh tangling:

Inversion and Overlap:

Preventing elements from inverting or overlapping during movement is essential. This requires strategies like regularization terms in the loss function to penalize negative volumes and ensure coherent adaptation.

Local and Global Coherence:
Ensuring local coherence in element displacement to prevent tangling, especially in regions with high gradient monitor functions, is challenging.

Suggestion:
Introduce a regularization term in the loss function to penalize negative volumes and use local displacement constraints to maintain element integrity.

D. Boundary conformance:

Complex boundaries:
Accurately adapting the mesh to complex geometries with sharp features and varying topologies without introducing distortions requires careful refinement near boundaries.

Boundary preservation:
Techniques such as boundary snapping and boundary layer refinement are necessary to maintain accurate geometric fidelity.

Suggestion:
For instance applying boundary snapping techniques and boundary layer refinement to ensure elements conform accurately to the physical boundaries.


E .Adaptive refinement and coarsening:

Dynamic adaptation:

Adapting the mesh based on evolving solution features, such as moving fronts, involves adding or removing elements dynamically.

Smooth transitions:

Ensuring smooth transitions between different resolution regions without creating poorly shaped elements is crucial. Hierarchical refinement and mesh smoothing techniques are often employed.



4/ Theoretical guarantees and limitations:

While the paper mentions that there are no theoretical guarantees that mesh tangling will be completely prevented, it would be beneficial to delve deeper into this limitation. Understanding the theoretical bounds of UM2N's performance and the conditions under which it might fail is important for setting realistic expectations.


5/ Real-world application scenarios:

Although the tsunami simulation is an excellent example, including more diverse real-world application scenarios would further demonstrate UM2N's versatility and robustness. For instance, applications in aerodynamics, biomechanics, or climate modeling could be explored.

Suggestion:

Add more case studies or at least detailed discussions of how UM2N could be applied to other real-world problems, emphasizing its practical benefits and potential challenges in these domains.


6/ Computational efficiency and scalability:

The paper discusses the reduction in computational costs compared to traditional methods but does not provide detailed benchmarks or discussions on the scalability of UM2N for very large meshes or real-time applications.

Suggestion:

Include more detailed benchmarks of computational efficiency and scalability, comparing UM2N's performance on large-scale problems with other methods.
Discuss strategies for optimizing performance and potential bottlenecks in the current implementation.


7/ Standpoint from Multi-Scale phenomena:

The paper does not address the specific challenges posed by multi-scale phenomena, where different parts of the domain may require vastly different resolutions to capture fine-scale features accurately while maintaining computational efficiency. Multi-scale phenomena often involve a wide range of spatial and temporal scales, making it difficult to balance resolution and computational cost effectively.

Suggestion:

Introduce experiments that specifically target multi-scale phenomena, demonstrating UM2N's ability to adapt the mesh dynamically to capture fine-scale features in high-gradient regions while coarsening in less critical areas. This could involve benchmarks on problems known for their multi-scale nature, such as turbulent flows or geophysical simulations. Additionally, discuss potential enhancements to the model, such as multi-grid techniques or hybrid approaches that combine UM2N with other multi-scale modeling strategies, to better handle these complex scenarios.

8/ Analysis from mesh continuity, local and global deformation viewpoints:

The paper does not thoroughly address how UM2N manages local and global deformations of the mesh, which are essential for accurately capturing complex physical phenomena. Effective mesh adaptation requires the ability to handle fine-scale local deformations to capture detailed features and large-scale global deformations to adapt to overall changes in the domain. Additionally, ensuring mesh continuity during these deformations is crucial to maintain the fidelity and stability of numerical simulations. Inadequate handling of these deformations can lead to poor resolution of critical areas, excessive computational costs, and potential discontinuities that degrade the accuracy of the simulation.

A/ Coupling local and global deformations:

Balancing local and global deformations is crucial for maintaining overall mesh quality and continuity. This involves ensuring that local refinements do not introduce excessive computational overhead and that global adaptations do not degrade the resolution of critical regions. Achieving this balance is technically challenging and requires sophisticated mesh adaptation algorithms that can handle both fine-scale and large-scale changes seamlessly.

B/ Mesh quality and continuity maintenance:

During both local and global deformations, maintaining high-quality elements (in terms of aspect ratio, skewness, and smoothness) is essential to avoid numerical instability and ensure accurate simulations. Poorly shaped elements can significantly degrade the performance of numerical solvers. Additionally, ensuring continuity in the mesh, where element shapes and sizes transition smoothly, is vital for maintaining numerical stability and accuracy.


9/ Analyze the optima of the learned meshes:

The paper does not address the issue of ensuring that the learned mesh configuration is globally optimal, nor does it consider the potential existence of multiple local optima in the optimization landscape. This oversight can result in suboptimal mesh configurations, which may not fully capture the desired features of the simulation domain or may introduce unnecessary computational overhead. In complex adaptive meshing scenarios, the risk of converging to local optima rather than the global optimum can significantly impact the accuracy and efficiency of the numerical solutions.

The optimization process for adaptive mesh movement often involves a highly non-convex loss landscape with many local minima. A single global optimum represents the best possible configuration of the mesh in terms of accuracy and computational efficiency. However, due to the complex nature of the problem, the optimization algorithm may converge to multiple local optima, each providing a suboptimal solution that fails to maximize the potential benefits of the adaptive meshing process.

The presence of multiple local optima can lead to inconsistent mesh configurations across different runs or simulations. This variability can make it difficult to ensure that the mesh is optimally adapted to the specific features of the simulation domain. Inconsistent convergence can also lead to variability in simulation results, reducing the reliability and robustness of the numerical methods.

Suggestion:

Benchmark the learned mesh against known optimal solutions or high-resolution reference meshes. This benchmarking can help assess how close the learned meshes are to the ideal configuration and identify specific areas where the model falls short.

**Questions:**

1/ How does UM2N handle extremely poor-quality initial meshes and highly irregular geometries?

Suggestion:

Conduct additional experiments with initial meshes characterized by highly skewed aspect ratios, large variations in element sizes, and complex boundary geometries. Include quantitative metrics on mesh quality before and after adaptation to highlight the robustness of UM2N.

2/  How does UM2N ensure mesh continuity and high-quality element shapes when dealing with both local and global mesh deformations?

3/ How does UM2N's learned mesh configuration compare to known optimal solutions or high-resolution reference meshes? Are there benchmarks or validation cases included to demonstrate the accuracy and efficiency of the mesh adaptation?

Suggestion:

Benchmark the learned mesh configurations against known optimal solutions or high-resolution reference meshes. Include comparisons using quantitative metrics such as error norms and gradient capture to highlight the accuracy and efficiency of UM2N in approaching the optimal mesh configuration.

4/ How does UM2N perform in resolving boundary layers and turbulent flows? Could you provide experimental results or benchmarks that validate its effectiveness in these critical fluid dynamics applications?

Suggestion:

Conduct targeted experiments on canonical boundary layer cases (e.g., flow over a flat plate) and turbulent flow benchmarks (e.g., turbulent channel flow). Use quantitative metrics like wall shear stress and turbulence intensity to evaluate performance and compare with high-fidelity simulations or experimental data.

5/  Can you elaborate on the rationale behind selecting Graph Attention Networks (GAT) over other Graph Neural Network (GNN) methods for mesh movement?
Additionally, have you considered or benchmarked against hierarchical GNNs, Octree-based methods, or point cloud-based methods, such as those utilizing multi-scale architectures like U-Net?
Exploring these alternatives could provide insights into potential improvements in handling mesh adaptation across different scales and complex geometries.

if the authors address the weaknesses and questions outlined above, I would be happy to increase my score. Doing so will significantly enhance the robustness and applicability of the proposed work.

**Limitations:**

Yes.

---

> ### Author Rebuttal · Authors · 2024-08-07
>
> We thank Reviewer yaVX providing a detailed summary of our strengths and valuable suggestions. Below, we present our detailed responses to the comments, indicating [W] for weaknesses and [Q] for questions.
>
> **Response to W1:**
> As the reviewer correctly identifies we do compare our method with existing methods, focussing on the M2N method. The reason for the limited number of comparisons is because there really has not been a lot of research so far on learning-based mesh-movement methods, most of the literature has focused on acceleration of other mesh adaptation strategies. We do cite the very recent flow2mesh paper by Jian Ju et al. (ref. [32] in our paper) who confirm our view: “However, these works all focused on mesh refinement rather than the mesh movement"
>
> **Response to W2:** We perform an additional ablation study of our network architecture on the Swirl case, analysing the contribution of each network component. We construct two models, which replace the GAT Decoder with one layer GAT denoted as UM2N-w/o-Decoder and remove the graph transformer denoted as UM2N-w/o-GT. The results are shown in Table R1. Both variants give worse results compared to the full UM2N. A visualization is also shown in Figure R2. It can be observed that without Decoder, the model distorts the shape of the ring, i.e., missing relative information between vertices; without the GT, the model fails to capture the details of the ring shape.
>
> **Response to W3/W4/Q1:**
> Thanks for raising this question. We include an additional experiment dealing with poor quality initial meshes as shown in Figure R4 and an additional experiment for irregular geometries. Also note the real-world Tsunami case shown in Figure 5 of our paper is also a good example for highly irregular geometries. And more cases with complex geometries are shown in our supplementary video.
> Figure R4 shows a low-quality input mesh with highly anisotropic elements. Using a monitor function based on the l2-norm of the gradient of the solution of an anisotropic Helmholtz problem, we tried applying conventional MA solvers and found that both the quasi-Newton and relaxation approaches failed to converge and/or resulted in tangled meshes, even though the elements are already aligned with the anisotropy of the Helmholtz solution. The UM2N approach, however, was able to successfully apply mesh movement without tangling.
> We would also like to clarify that our work targets mesh movement that dynamically adapts to a PDE solution, which is not a replacement for mesh generation in general, so we would always expect a reasonable input mesh when applying our model.
> In Figure R1, we show a stress test for both MA and UM2N. It can be observed that MA soon fails in the presence of non-convexity in the geometry while UM2N only gets tangled in the most challenging right angle case. It indicates that our method allows a relaxation of the geometric limitations compared to MA, although we admit that there  is no theoretical guarantee for no mesh tangling as we mentioned in the limitation.
>
> **Response to W5/W6:**
> We appreciate that the reviewer acknowledges that our Tsunami case is an excellent real-world scenario application. We also kindly remind that there are also more applications shown in our supplementary video. We do agree more examples for different areas would better demonstrate the versatility of UM2N. Due to the time limit in the rebuttal period, we would like to leave this to our revisions.
> For some limited scalability, we further apply our trained model on a flow past cylinder case with ~11k vertices and ~22k triangles as shown in Figure R5. It is shown that our UM2N works well on this case with more degrees of freedom compared to the one in the paper.
> For an even larger case, we perform stress tests using a time-independent Helmholtz case on our RTX 3090 24 GB GPU and observe that the maximum scale of the problem that can be run on this GPU has ~50k number of elements. The inference time is ~760 ms, the MA method on the same problem requires ~37800 ms with a residual threshold 1e-4. Considering this limitation, applying the memory efficient transformer (linear-attention transformer for further improvement of the inference efficiency) is targeted as future work.
>
> **Response to Q2/W8:** There are two types of mesh continuity to consider: continuity in space (i.e. mesh smoothness) and continuity in time (avoiding large changes between timesteps). The latter is promoted by training with samples based on  Monge Ampere solutions. Based on optimal transport theory, these represent the unique minimal change transformation that satisfies the equidistribution principle provided certain smoothness and convexity criteria are satisfied. Given that this is already the unique optimal mesh that with these properties, quality of elements in terms of their aspect ratio is not explicitly enforced, but the minimal change property does mean a high quality input mesh is transformed to a close-in-quality output mesh as long as the required transformation in terms of the monitor function is not too demanding (e.g. non-smooth). It should be noted it is not the case that equal-aspect triangles always provide the highest quality mesh. This in fact depends on the anisotropy of the PDE solution, and in cases where more control over the aspect ratio is required, other mesh movement methods allow for a tensorial input metric, but this necessarily means the minimal change (continuity) property is no longer satisfied in general. There is thus always a trade-off between different desired properties. Likewise, when asking for a considerable global redistribution of degrees of freedom this may come at the cost of the local mesh quality. Any mesh movement method will therefore be a compromise with UM2N focusing on maintaining minimal movement and maintaining desired output cells areas.
>
> Please refer to **Supplement Response to yaVX** in general response for more responses **Q3/W9, Q4, Q5**.

---

> > ### Comment · Reviewer_yaVX · 2024-08-09
> > **Response to the rebuttal and follow-up**
> >
> > Thank you for providing detailed responses to my comments and questions. I appreciate your efforts to address the points raised and for conducting additional experiments to clarify the strengths and limitations of the Universal Mesh Movement Network (UM2N). Here are some additional thoughts.
> >
> > **Comparison with State-of-the-art methods:**
> >
> > I understand the challenge in comparing UM2N with a broader set of state-of-the-art methods due to the limited number of learning-based mesh movement approaches. Your focus on the M2N method and the confirmation from the Flow2Mesh paper about the scarcity of learning-based mesh movement methods is noted. However, as the field evolves, it might be beneficial to continually update your comparisons with new emerging methods. Exploring the literature for recent advancements and including these in future versions of the paper would be valuable.
> >
> > **Ablation studies:**
> >
> > Thank you for conducting additional ablation studies. The results provided in Table R1 and Figure R2 help elucidate the contributions of each network component. It would be interesting to see a more detailed analysis of how different hyperparameters impact the performance of UM2N, which could be explored in future work.
> >
> > **Handling poor-quality meshes and irregular geometries:**
> >
> > Your additional experiments on poor-quality initial meshes and irregular geometries, including the real-world tsunami case and stress tests, effectively demonstrate UM2N's robustness. The clarification that UM2N targets mesh movement rather than mesh generation is helpful. It’s commendable that UM2N performs well even with challenging input meshes. However, it would be beneficial to include detailed quantitative metrics in future revisions to further emphasize UM2N's capabilities.
> >
> > **Computational efficiency and scalability:**
> >
> > The scalability experiments and the comparison with the MA method are informative. It’s promising to hear about future work involving memory-efficient transformers for improved inference efficiency. Continued focus on optimizing computational performance and discussing potential bottlenecks will be crucial as UM2N scales to even larger and more complex problems.
> >
> > **Mesh continuity and quality:**
> >
> > Your explanation of the trade-offs between mesh quality, continuity, and the minimal movement property is insightful. It provides a clear rationale for UM2N's design choices. Future work could explore strategies for balancing these trade-offs, especially in scenarios requiring significant global redistributions.
> >
> > **Optimal mesh configurations:**
> >
> > I appreciate your discussion on the challenges of proving mesh optimality and the use of PDE solution error reduction rate as a metric. The references to high-resolution meshes and your detailed error analysis are compelling. Benchmarking against known optimal solutions or employing more sophisticated error metrics could further strengthen these comparisons in future work.
> >
> > **Performance in boundary layers and turbulent flows:**
> >
> > The experiments on flow past parallel plates provide valuable insights into UM2N’s capability to resolve boundary layers. Continued exploration of turbulent flow scenarios, along with detailed performance metrics, would further solidify UM2N’s applicability in critical fluid dynamics applications.

---

> > > ### Comment · Reviewer_yaVX · 2024-08-09
> > > **Follow-up questions**
> > >
> > > 1. In future revisions, could you provide more detailed quantitative metrics to highlight UM2N's performance, especially concerning mesh quality and computational efficiency?
> > > 2. Could you discuss any specific strategies you are considering for optimizing UM2N’s performance on large-scale problems, beyond memory-efficient transformers?
> > > 3. Given the trade-offs between mesh quality, continuity, and minimal movement, are there any specific scenarios where you prioritize one over the others? How do you envision addressing these trade-offs in future work?
> > > 4. Regarding the temporal continuity of meshes, how does UM2N ensure smooth transitions between time steps, especially in dynamic simulations? Are there any specific temporal regularization techniques or loss functions applied during training?

---

> > > > ### Author Response · Authors · 2024-08-12
> > > > **Response to follow-up questions and thank you**
> > > >
> > > > We thank for the reviewer's constructive feedback and acknowledgment of our efforts to address the concerns. We are very happy and greatly appreciate that the reviewer improve the score!
> > > >
> > > > Here are our responses to the follow-up questions:
> > > >
> > > > **Response to Q1:** Thanks for the suggestion. We would commit to provide more quantitative metrics for the UM2N for the mesh quality and computational efficiency in the revision.
> > > >
> > > > **Response to Q2:** Thanks for raising this question. For large-scale problems, in addition to applying memory-efficient transformers, the computation efficiency is another issue as the time-complexity of self-attention is O(N^2) where N is the number of vertices of a mesh.
> > > > The main strategy is to apply linear-attention transformers. The linear-attention reduces the time-complexity  from O(N^2)  to O(N). There are many existing works in this direction [1][2] from the natural language processing community as well as in the neural PDE solver community [3][4]. In addition, attention with masks may also help improve the efficiency. Similar intuitions from [5], we hypothesize there is redundant information in a representation of meshes and masking would not severely degrade the performance. These preliminary ideas for handling large-scale problems can be explored in the future work.
> > > >
> > > > [1] Katharopoulos, A. et al, Transformers are RNNs: Fast Autoregressive Transformers with Linear Attention (ICML 2020)
> > > >
> > > > [2]Kitaev et al, Reformer: The Efficient Transformer (ICLR 2020)
> > > >
> > > > [3] Cao et al, Choose a Transformer: Fourier or Galerkin (NeurIPS 2021)
> > > >
> > > > [4] Hao et al, GNOT: A General Neural Operator Transformer for Operator Learning (ICML 2023)
> > > >
> > > > [5] He et al, Masked Autoencoders Are Scalable Vision Learners (CVPR 2022)
> > > >
> > > > **Response to Q3:** This is a great question that touches on important aspects of the paper. One distinction in types of scenarios, is on the one hand steady state problems, or problems where one doesn't attempt to dynamically follow time-dependent features with mesh resolution, which focus on the optimisation of a single mesh and its spatial properties, and unsteady problems where the mesh is adapted in time to focus resolution on time-dependent features or to adjust to moving boundaries. For the second category minimizing the amount of change between subsequent meshes, as you mention, becomes very important. One important practical aspect in this category, which we didn't really get the chance to discuss in the paper, is that of solution transfer between subsequent meshes, e.g. through interpolation which introduces artificial diffusion and which gets exacerbated by "unnecessary" mesh changes. Another approach, the so called Arbitrary Lagrangian Eulerian (ALE) method, does not interpolate the solution but allows the solution to "move with the mesh" which is accommodated by evaluating the PDE in a moving reference frame defined by the mesh velocity derived from the change  between subsequent meshes. This method can reduce both the amount of diffusion and costs associated with interpolation, but this is again limited by the amount of change in each timestep which can be related to a CFL-like condition. Focusing on this aspect of continuity between meshes there can indeed be a trade-off with the spatial, instantaneous mesh quality properties of each mesh individually. We believe this trade off should also be seen in light of the core assumption of mesh movement methods, which is that the topology of the mesh is unchanged. If flow features are followed slowly over time, the twisting of these features may lead to inevitable tangling of the mesh structure, even if large movements are allowed, which may only be resolved through complete remeshing, altering the connectivity of the mesh. The same problem arises with moving-boundary problems, e.g. the flow around rotating structures. For
> > > > future directions, we are therefore very much interested in hybrid approaches that combine r and h adaptivity, allowing for pure mesh movement in most timesteps but intervening with topological operation when the quality of the mesh degrades to much.
> > > >
> > > > **Response to Q4:** There are no explicit terms during the training promoting mesh continuity other than our use of the Monge Ampere solution as the "optimal" reference mesh. The minimal change property is in a sense the defining feature of the Monge-Ampere approach, and using it to move meshes in time-dependent problems indeed results in smoothly varying meshes over time provided the changes in the monitor function are also smooth. Although there is no guarantee, we have observed the same smoothness properties and not seen any timestep-to-timestep back-and-forth changes that might be expected when approximating the movement induced by the monitor function.

---

> > > > > ### Comment · Reviewer_yaVX · 2024-08-13
> > > > > **Thank you**
> > > > >
> > > > > Thank you for the detailed responses. Please include the discussion in the revised paper

---

> > > > > > ### Author Response · Authors · 2024-08-13
> > > > > >
> > > > > > Thanks again for your comments and engaging dialogue. We commit to include the discussion into the revision.

---

> ### Comment · Reviewer_yaVX · 2024-08-09
> **Summary**
>
> The authors have provided a comprehensive and thoughtful rebuttal, effectively addressing the primary concerns and questions raised about the UM2N architecture. They clarified their focus on mesh movement rather than generation and highlighted the challenges of comparing their work with the limited number of learning-based mesh movement methods available. The additional experiments conducted, especially on poor-quality initial meshes and irregular geometries, demonstrate UM2N’s robustness and versatility. Their detailed responses on the integration of Graph Transformer and Graph Attention Network (GAT) components, the role of element volume loss, and handling mesh deformations provide valuable insights into the technical strengths of their approach. However, as the field evolves, ongoing comparisons with new methods, further exploration of hyperparameter sensitivities, and expansion of real-world applications will strengthen the work’s impact and applicability.
>
> Overall, the authors have made a strong case for the effectiveness and innovation of UM2N, particularly through their emphasis on zero-shot generalization and computational efficiency. The clear explanations and additional experiments help illustrate the model's capabilities in diverse scenarios. I appreciate the thoroughness and clarity of the paper, as well as the authors' proactive engagement with the feedback. The improvements and insightful discussions provided in the rebuttal demonstrate the paper's potential to advance simulation technologies through adaptive mesh movement.

---

### Official Review · Reviewer_xMNp · 2024-07-22

**Soundness:** 2
**Presentation:** 4
**Contribution:** 4
**Rating:** 5
**Confidence:** 4

**Summary:**

In the present work, authors tackle the challenging task of accelerating PDE-solving process with deep learning, focusing on mesh movement problem. They suggest a new architecture that combines graph transformer encoder and graph attention decoder to significantly accelerate the PDE-solving. Authors propose a way to decouple the underlying PDE solving from the mesh movement process itself. It makes the approach universal with regard to solvers as well as to boundary geometries and mesh structures. They demonstrate the superiority of their method over conventional solver, and the ability on various benchmark tasks.

**Strengths:**

The work is well written and concise. The work focuses specifically on the r-adaptation, i.e. mesh movement problem. Authors carefully describe what the complete problem is and which domain their method attributes to. This might look excessive but overall leaves good impression and facilitates understanding.

The approach itself of generating the dataset of random fields and train the model to learn an auxiliary PDE sounds compelling and indeed universal. It allows them to effectively decouple the monitor values from the PDE solved, which is one of the main strengths of the work.

The use of data, the architecture, the losses used and other deep learning-related details are well explained; the losses and the input varioations are justified in the convincing ablation study.

Authors claim that they would like to provide the code for reproducibility, which is always for the better.

**Weaknesses:**

Authors do not demonstrate a single experiment, where their model provides mesh tangling. Authors explain such effect by the use of the volume loss (line 326) calling it "tangling-aware". In general, it might be good, since one can be sure that any use of UM2N is reliable.
Yet, without proper evaluation (or at least, clarification for such perfection), It cannot be regarded as a virtue. It implies either that the experiment setups are chosen to be weak or the model has the inductive bias that might limit its universality. In other words, there might exist some setups that the model won't be able to estimate correctly, because of oversmoothness. As for example, one can imagine the domain with a tight bottleneck, the flow might not propagate to the other side at all.

Another side of the same issue can be seen in the flow-past-cylinder experiment. Both MA and M2N provide some tangling and hence considered as "Failed", so entire discussion in this experiment is in favor of the proposed UM2N. Authors call the setup "classic and challenging", yet there is not a single baseline that provides at least some solution. Either another baseline that actually gives some number should be provided or the setup should be simplified, e.g. choosing the object that provides less turbulence, such as the airfoil.

**Questions:**

I would appreciate if authors discuss some of the issues I raised above in the Weaknesses.

Mainly: non-tangled results of other methods on the flow-past-cylinder; tangled results of the UM2N model itself

**Limitations:**

-

---

> ### Author Rebuttal · Authors · 2024-08-07
>
> We like to thank the reviewer for their kind words on the presentation and level of contribution of our paper. Below, we present our detailed responses to the comments, indicating **[W]** for weaknesses and **[Q]** for questions.
>
> We appreciate the feedback regarding the amount of explanation we were able to provide regarding limitations of our approach and validation of the flow past a cylinder case. As the reviewer correctly identifies our paper focuses specifically on r-adaptation which has benefits over other mesh adaptation methods - maintaining mesh topology, and, in the Monge-Ampere (MA) approach, minimising the amount of change - but also drawbacks: solving the MA equations is very costly, and it puts limitations on the physical geometry, demanding smooth boundaries and convex domains. In our paper, we demonstrate a significant reduction in the cost through the UM2N acceleration. We also demonstrate a relaxation to some extent of the geometric limitations, but this is indeed a bit harder to outline and quantify.
>
> **Response to W1/Q1:‘tangled results of the UM2N model’:** There are in fact still cases where UM2N might fail, we perform additional experiments as shown in Figure R1. Starting from a rectangle, we gradually distort the geometry to a more non-convex shape i.e., the case is more and more challenging from top to bottom. The UM2N finally fails in the case surrounding sudden jumps in the boundary accompanied with a large variation in the required resolution as shown in the bottom row of Figure R1: a sudden constriction of the flow leads to tangling with UM2N in the two left corners of the channel. In a more smoothed version UM2N does produce a valid mesh, where the original MA method still fails (second to bottom rows in Figure R1). We agree that this warrants a more extensive discussion in our paper in the revision.
>
> **Response to W2/Q2: ‘non-tangled results of other methods on the flow-past-cylinder’:**
> We additionally perform a non-tangled flow-past-cylinder shown in Figure.R6 as mentioned by the reviewer xMNp. We simplified the setting: lower the Rrenynolds number by reducing the inflow velocity and set the top and bottom slip boundary, which finally makes it a laminar flow. As shown in Figure.R6, both the UM2N and M2N give non-tangled meshes and it can be observed that UM2N better captures the dynamics i.e., adapts to the PDE solution of stable state compared to the High Res solution. The quantitative results shown in the Tab.R3 also indicate that UM2N perform better than M2N regarding to error reduction.
>
>
> The aim of the flow past a cylinder case was to demonstrate that mesh movement can now readily be applied, which is not feasible with a classical approach like MA for several reasons: the non-convexity of the domain means the equations are fundamentally ill-posed, and, even if that could be overcome, e.g. by applying mesh movement only in the right half of the domain, the prohibitive costs to apply this every timestep. This case is based on the so called DFG-benchmark 2D-2 [1], which has been modelled with a variety of other frameworks, including those that incorporate other mesh adaptation approaches [see e.g. 2-5] The results in our paper were validated by comparison with a high resolution uniform-mesh model, which corresponds well with the reference value in [2,3]. Note that convergence to the reference depends on various discretisation choices, finite-element pair, type of boundary condition, etc., which are not the focus of our paper. We therefore decided to only compare with a high resolution case, using the same discretisation details, to demonstrate the improvement in accuracy through mesh movement alone. We would like to clarify this however in our paper, and make better reference to and comparison with published results for this case.
>
>
> [1] https://wwwold.mathematik.tu-dortmund.de/~featflow/en/benchmarks/cfdbenchmarking/flow/dfg_benchmark2_re100.html
>
> [2] D. Capatina et al. ‘22 https://doi.org/10.1016/j.cma.2019.112775
>
> [3] V. John ‘04 https://onlinelibrary.wiley.com/doi/abs/10.1002/fld.679
>
> [4] T. Coupez et al. ‘13 https://doi.org/10.1016/j.jcp.2012.12.010
>
> [5] Hachem et al ‘13 https://doi.org/10.1002/nme.4481

---

> > ### Comment · Reviewer_xMNp · 2024-08-09
> > **reply to Rebuttal**
> >
> > I thank authors for their clear and instructive reply to all the issues I raised.
> >
> > Authors added new experiments and showed where their method fails and provides mesh tangling.
> >
> > Also, they showed non-tangled results for flow-over-cylinder for the MA method. I believe this was not very informative when the proposed method seems to perform fine, while others completely fail (even if this fail is inherent). Authors found where this fail boundary lies and illustrated a more compelling comparison; this definitely improves the overall picture.

---

> > > ### Author Response · Authors · 2024-08-09
> > > **Thank you!**
> > >
> > > Dear Reviewer xMNp,
> > >
> > > Thanks again for the time and effort you've invested in reviewing our manuscript. We also greatly appreciate your acknowledgment of our efforts to address your concerns.
> > >
> > > Your insights have been instrumental in improving our work, and we are committed to incorporating your suggestions into the revision. We would also greatly appreciate your consideration in possibly raising the original rating if you find our responses satisfactory. If you believe there are any remaining areas where additional clarifications/responses could contribute to such a higher rating, please don't hesitate to inform us. Your guidance is always crucial to improve the quality of our submission.
> > >
> > > Thank you very much,
> > >
> > > Authors

---

### Author Rebuttal · Authors · 2024-08-07

We are glad to receive valuable and constructive comments from all the reviewers. We have made a substantial effort to clarify reviewers' doubts and enrich our experiments in the rebuttal phase. In our responses, Tab. Rxx or Fig. Rxx
refers to the new Rebuttal results in the attached PDF. Below is a summary of our responses:

**Reviewer xMNp:**
1. We perform additional experiments to demonstrate a failing case for UM2N and to explore the limits to its ability to avoid mesh tangling (Figure. R1)
2. We demonstrate how UM2N provides a relaxation to some extent of the geometric limitations compared to the Monge-Ampère (MA) based method. (Figure. R1)
3. We conduct additional experiments to show non-tangled results of other methods on the flow-past-cylinder (Figure. R6), and provide an explanation on why MA inherently fails on flow-past-cylinder.

**Reviewer yaVX:**
1. We perform an additional ablation study on network components (Table.R1/Figure.R2) to demonstrate the effectiveness of both Graph Transformer based encoder and Graph Attention Network based deformer.
2. We conduct an additional experiment dealing with poor quality initial meshes (Figure R4).
3. We provide a detailed discussion about mesh continuity in mesh movement.
4. We provide a clarification for optimality of mesh movement and explain our decision on the metric we use for evaluating mesh movement. We also provide measures between model output mesh and target mesh. (Tab. R2)
5. We perform an additional boundary layer experiment. (Figure.R3)
6. We explain the rationale for selecting Graph Transformer and Graph Attention Network and discuss the other related networks as potential future work.

**Supplement Response to yaVX:**

**Response to Q3/W9:**
Here we would like to discuss both the comparisons to optimal mesh and reference PDE solutions. It is generally hard to strictly prove the optimality of a mesh in cases with more than one dimension, and, as argued previously, there are various definitions of optimal that can be considered, and, in a sense, the meshes obtained from the MA method can be regarded as optimal meshes already. In Table R2, we provide the coordinate loss, element volume loss and Chamfer distance during training. Note that these metrics for measuring the difference between model output mesh and MA reference mesh are intermediates. The ultimate metric we care about is the PDE solution error reduction rate (ER), which is why we focus on ER in our paper.

We provided reference solutions for both Swirl and Flow past cylinder cases in our paper which are computed on high-resolution meshes (~10k for swirl, ~260k for flow-past-cylinder), indicated as ‘High Res’. As shown in Figures 2 and 3 of the paper, the solution obtained on a UM2N moved mesh has a smaller L2 error (plots in right part of Figure 2) and more accurate drag coefficient (bottom part of Figure 3) compared to that obtained on an original uniform mesh. These errors are computed against the reference, high-res PDE solution.

**Response to Q4:**  In our flow past the cylinder case, at the top and bottom we imposed non-slip boundary conditions which already promotes a boundary layer similar to the suggested flow over a flat plate.
For a clearer illustration, we have now additionally conducted a flow past two parallel plates experiment. We observe that the velocity parallel to the boundary ($u_y$) obtained from UM2N moved mesh better aligns with the high resolution results compared to the results on uniform mesh as shown in the Figure R3.

**Response to Q5:** Thank you for this inspiring question. Graphs are a natural way to represent unstructured meshes. We apply the graph transformer as encoder for its powerful expressivity, and apply the graph attention network as decoder because it can naturally help alleviate mesh tangling with some extra design. More specifically, we update the coordinate of each vertex with the combination of its neighbours, the coefficients of which are determined by the GAT module. This guarantees each vertex to only move within the convex hull of its neighbouring vertices, hence effectively alleviating mesh tangling issues.

We thank the reviewer for their suggestions for other benchmark methods. We agree these methods have their strengths but not all are equally relevant to our fully unstructured mesh based approach. For example, octree-based methods can be only applied to octree-based hierarchical meshes; point-cloud-based models ignore the topological relationship between vertices. Hierarchical models or multi-scale architectures such as U-Net is a very interesting idea, as the input and output of the model share the same structure, and capturing and sharing local and global information with hierarchical model structures is appealing, hence is definitely worth further investigation. In general, we agree with the reviewer that exploring these methods for mesh adaptation are interesting directions for future work.

**Reviewer jEN5:**
1. We perform an additional ablation study on network components (Table.R1/Figure.R2) to demonstrate the effectiveness of both Graph Transformer based encoder and Graph Attention Network based deformer.
2. We compare our method to related work qualitative and discuss the potential future direction.
3. We discuss the intuition and limitation of our model handing boundary with complex geometries.
4. We conduct additional experiment on a larger case (Figure.R5) and investigate the scalability based on our computational resources.

**Reviewer Jzyt:**
1. We provided clarification about the computation of monitor values as well as our pipeline.
2. We provide explanation about the chamfer distance issue the reviewer concerns.
3. We conduct an additional experiment dealing with poor quality initial meshes and visualize the results. (Figure R4).

---

### Decision · Program_Chairs · 2024-09-25

**Decision:**

Accept (spotlight)

**Comment:**

Initially all reviewers gave a weak or borderline accept rating. The authors wrote a strong rebuttal that convinced two of the reviewers to upgrade to strong accept with the third one maintaining his weak accept.